# Harmonizing the object recognition strategies of deep neural networks with humans

**Thomas Fel**[*1,2], **Ivan Felipe**[*1], **Drew Linsley**[*1,3], **Thomas Serre**[1,2,3]
{thomas_fel,ivan_felipe_rodriguez,drew_linsley}@brown.edu

## Abstract

The many successes of deep neural networks (DNNs) over the past decade have largely been driven by computational scale rather than insights from biological intelligence. Here, we explore if these trends have also carried concomitant improvements in explaining the visual strategies humans rely on for object recognition. We do this by comparing two related but distinct properties of visual strategies in humans and DNNs: *where* they believe important visual features are in images and *how* they use those features to categorize objects. Across 84 different DNNs trained on ImageNet and three independent datasets measuring the *where* and the *how* of human visual strategies for object recognition on those images, we find a systematic trade-off between DNN categorization accuracy and alignment with human visual strategies for object recognition. *State-of-the-art DNNs are progressively becoming less aligned with humans as their accuracy improves.* We rectify this growing issue with our neural harmonizer: a general-purpose training routine that both aligns DNN and human visual strategies and improves categorization accuracy. Our work represents the first demonstration that the scaling laws [1–3] that are guiding the design of DNNs today have also produced worse models of human vision. We release our code and data at `https://serre-lab.github.io/Harmonization` to help the field build more human-like DNNs.

## 1 Introduction

Rich Sutton stated [4] that the bitter lesson "from 70 years of AI research is that general methods that leverage computation are ultimately the most effective, and by a large margin." Deep learning has been the standard approach to object categorization problems ever since the paradigm shifting success of AlexNet [5] on the ImageNet [6] benchmark a decade ago. As deep neural network (DNN) performance has continued to improve in the intervening years, Sutton's lesson has become more fitting than ever, with recent networks rivaling and likely outperforming humans on the benchmark [7] through brute-force computational scale: increasing the number of network parameters and number of images used for training orders-of-magnitude beyond AlexNet [1–3]. While the successes of so-called "scaling laws" are undeniable, this singular focus on performance in the field has side-stepped an equally important question that will govern the utility of object recognition models for the brain sciences and industry applications alike: *are the visual strategies learned by DNNs aligned with those used by humans?*

The visual strategies that mediate object recognition in humans can be decomposed into two related but distinct processes: identifying *where* the important features for object recognition are in a scene, and determining *how* to integrate the selected features into a categorical decision [8, 9]. It has been

---

[*]These authors contributed equally.
[1]Department of Cognitive, Linguistic, & Psychological Sciences, Brown University, Providence, RI
[2]Artificial and Natural Intelligence Toulouse Institute (ANITI), Toulouse, France
[3]Carney Institute for Brain Science, Brown University, Providence, RI

36th Conference on Neural Information Processing Systems (NeurIPS 2022).

known for nearly a century [10–13] that different humans attend to similar locations when asked to find and recognize objects. After selecting these important features, human observers are also consistent in how they use those features to categorize objects – the inclusion of a few pixels in an image can be the difference between recognizing an object or not [9, 14].

Has the past decade of DNN development produced any models that are aligned with these human visual strategies for object recognition? Such a model could transform cognitive science by supporting a better mechanistic understanding of how vision works. More human-like models of object recognition would also resolve the problems with predictablity and interpretablity of DNNs [15–18], and control their alarming tendency to rely on "shortcuts" and dataset biases to perform well on tasks [19]. In this work, we perform the first large-scale and systematic comparison of the visual strategies of DNNs and humans for object recognition on ImageNet.

**Contributions.** In order to compare human and DNN visual strategies, we first turn to the human feature importance maps collected by Linsley et al. [20, 21]. Their datasets, *ClickMe* and *Clicktionary*, contain maps of nearly 200,000 unique images in ImageNet that highlight the visual features humans believe are important for recognizing them. These datasets amount to a reverse inference on *where* important visual features are in ImageNet images (Fig. 1). We complement these datasets with new psychophysics experiments that directly test how important visual features are used for object recognition (Fig. 1). **As DNN performance has increased on ImageNet, their alignment with human visual strategies captured in these datasets has worsened.** This trade-off is found over 84 different DNNs representing all popular model classes – from those trained for adversarial robustness to those pushing the scaling laws in network capacity and training data. To summarize our findings:

- The trade-off between DNN object recognition accuracy and alignment with human visual strategies replicates across three unique datasets: *ClickMe* [20], *Clicktionary* [21], and our psychophysics experiments.

- We shift this trade-off with our neural harmonizer, a novel drop-in module for co-training any DNN to align with human visual strategies while also achieving high task accuracy. Harmonized DNNs learn visual strategies that are significantly more aligned with humans *than any other DNN we tested*.

- We release our data and code at `https://serre-lab.github.io/Harmonization/` to help the field tackle the growing misalignment between DNNs and humans.

## 2  Related work

**Do DNNs explain human visual perception?** Despite the continued success of DNNs on computer vision benchmarks, there are conflicting accounts on their ability to explain human vision. On the one hand, there is evidence that DNNs are improving as models of human visual perception on challenging tasks, such as recognizing objects obscured by noise [23]. On the other hand, there is also evidence that DNNs struggle to explain perceptual phenomena in human vision like contextual illusions [24], perceptual grouping [19, 25, 26], and categorical prototypes [27]. Others have found

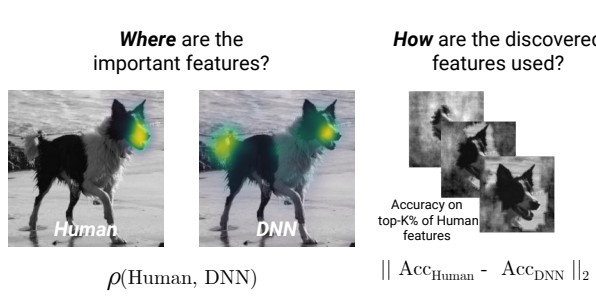

Figure 1: **Visual strategies of object recognition**. We investigate the alignment of human and DNN visual strategies in object categorization. We decompose human visual strategies into descriptions of *where* important features are [20, 22], and *how* those features are integrated into visual decisions.

differences between human attention data and DNN models of visual attention [20, 28]. Moreover, DNNs have stopped improving as models of the ventral visual system in humans and primates over recent years. While the original theory was that model explanations of object-evoked neural activity patterns improved alongside model categorization accuracy [29], recent large-scale DNNs are worse at explaining neural data than older ones with lower ImageNet accuracy [30].

**What are the visual strategies underlying human object recognition?** Ever since its inception, a goal of vision science has been to characterize the neural processes supporting object recognition in humans. It has been discovered that object recognition can be decomposed into different processing stages that emerge over time [8, 31–36], where the earliest stage is associated with processing through feedforward connections in the visual system, and the later stage is associated with processing through feedback connections. Since the DNNs used today mostly rely on feedforward connections, it is likely that they are better models for that rapid feedforward phase of processing than the subsequent feedback phase [33, 37]. To maximize the likelihood that the visual strategies learned by DNNs align with those used by humans, our experiments focus on the visual strategies of rapid feedforward object recognition in humans.

Most closely related to our work, are studies of "top-down" image saliency and *where* category diagnostic visual features are in images. These studies typically involve asking participants to search for an object in an image, or find visual features that are diagnostic for an object's category or identity [10–13, 20, 22, 38]. In our work, we complement these descriptions of *where* important features are in images with psychophysics testing *how* those features are used to categorize objects.

**Comparing visual strategies of humans and machines.** As methods in explainable artificial intelligence have developed over the past decade, they have opened up opportunities for comparing the visual regions selected by humans and DNNs when solving tasks. Many of these comparisons have focused on human image saliency measurements captured by eye tracking or mouse clicks during passive or active viewing [20, 22, 39–42]. Others have compared categorical representation distances [40, 43] or combined those distances with measures of human attention [28, 44]. The most direct comparisons between human and DNN visual strategies involved analyzing the minimal image patches needed to recognize objects [9, 45, 46]. However, these studies were limited and compared humans with older DNNs on tens of images. To the best of our knowledge, the largest-scale evaluation of human and DNN visual strategies relied on the *ClickMe* dataset to compare visual regions preferred by humans and attention models trained for object recognition [20]. What is noticeably missing from each of these studies is an large-scale analysis spanning many images and models of how human and DNN alignment has changed as a function of model performance.

**Improving the correspondence between humans and machines.** Inconsistencies between human and DNN representations can be resolved by directly training models to act more like humans. DNNs have been trained to have more human-like attention, or human-like representational distances in their output layers [20, 40, 43, 47, 48]. Here, we add to these successes with the neural harmonizer, a training routine that automatically aligns the visual strategies (Fig. 1) of any two observers by minimizing the dissimilarity of their decision explanations.

## 3 Methods

**Human feature importance datasets.** We focused on the ImageNet dataset to compare the visual strategies of humans and DNNs for object recognition at scale. We relied on the two significant efforts for gathering feature importance data from humans on ImageNet: the *Clicktionary* [22] and *ClickMe* [20] games, which use slightly different methods to collect their data. Both games begin with the same basic setup: two players work together to locate features in an object image that they believe are important for categorizing it. As one of the players selects important image regions, those regions are filled into a blank canvas for the other observer to see and categorize the image as quickly as possible. In *Clicktionary* [22], both players are humans, whereas in *ClickMe* [20], the player selecting features is a human and the player recognizing images is a DNN (VGG16 [49]). For both games, feature importance maps depicting the average object category diagnosticity of every pixel was computed as the probability of it being clicked by a participant. In total, *Clicktionary* [22] contained feature importance maps for 200 images from the ImageNet validation set, whereas *ClickMe* [20] contained feature importance maps for a non-overlapping set of 196,499 images from ImageNet training and validation sets. Thus, *ClickMe* has far more data than *Clicktionary*, but *Clicktionary* data has more reliable human feature importance data than *ClickMe*. Our experiments measure the alignment between human and DNN visual strategies using *ClickMe* and *Clicktionary* feature importance maps captured on the ImageNet validation set. As we describe in §4, *ClickMe* feature importance maps from the ImageNet training set are used to implement our neural harmonizer.

**Psychophysics participants and dataset.**   We complemented the feature importance maps from *Clicktionary* and *ClickMe* with psychophysics experiments on rapid visual categorization. We recruited 199 participants from Amazon Mechanical Turk (`mturk.com`) to complete the experiments. Participants viewed a psychophysics dataset consisting of the 100 animal and 100 non-animal images in the Clicktionary game taken from the ImageNet validation set [22]. We used the feature importance maps for each image as masks for the object images, allowing us to control the proportion of important features observers were shown when asked to recognize objects (Fig. 5a). We generated versions of each image that reveal anywhere between 1% to 100% (at log-scale spaced intervals) of the important object pixels against a phase scrambled noise background (see Appendix §1 for details on mask generation). The total number of revealed pixels was equal for every image at a given level of image masking, and the revealed pixels were centered against the noise background. Each participant saw only one masked version of each object image.

**Psychophysics experiment.**   Participants were instructed to categorize images in the psychophysics dataset as animals or non-animals as quickly and accurately as possible. Each experimental trial consisted of the following sequence of events overlaid onto a white background (SI Fig. 1): (*i*) a fixation cross displayed for a variable time (1,100–1,600ms); (*ii*) an image for 400ms; (*iii*) an additional 150ms of response time. In other words, the experiment forced participants to perform rapid object categorization. They were given a total of 550ms to view an image and press a button to indicate its category (feedback was provided on trials in which responses were not provided within this time limit). Images were sized at 256 x 256 pixel resolution, which is equivalent to a stimulus size approximately between 5 – 11 degrees of visual angle across a likely range of possible display and seating setups we expect participants used for the experiment. Similar paradigms and timing parameters have been shown to capture pre-attentive visual system processing [31, 50–52]. Participants provided informed consent electronically and were compensated $3.00 for their time ($\sim$ 10–15 min; approximately $15.00/hr).

**Models.**   We compared humans with 84 different DNNs representing the variety of approaches used in the field today: 50 CNNs trained on ImageNet [1, 49, 53–64, 64–73], 6 CNNs trained on other datasets in addition to ImageNet (which we refer to as "CNN extra data") [1, 66, 74], 10 vision transformers [75–79], 6 CNNs trained with self-supervision [80, 81], and 13 models trained for robustness to noise or adversarial examples [82, 83]. We used pretrained weights for each of these models supplied by their authors, with a variety of licenses (detailed in SI §2), implemented in Tensorflow 2.0, Keras, or PyTorch.

# 4   Results

## 4.1   *Where* are diagnostic object features for humans and DNNs?

To systematically compare the visual strategies of object recognition for humans and DNNs on ImageNet, we first turned to the *ClickMe* dataset of feature importance maps [20]. In order to derive comparable feature importance maps for DNNs, we needed a method that could be efficiently and consistently applied to each of the 84 DNNs we tested without any idiosyncratic hyperparameters. This led us to choose a classic method for explainable artificial intelligence, image feature saliency [84]. We prepared human feature importance maps from *ClickMe* by taking the average importance map produced by humans for every image that also appeared in ImageNet validation. We then used Spearman's rank-correlation to measure the similarity between human feature maps and DNN feature maps for each image [50]. We also computed the inter-rater alignment of human feature importance maps as the mean split-half correlation across 1000 random splits of the participant pool ($\rho = 0.66$). We then normalized each human-DNN correlation by this score [20].

There were dramatic qualitative differences between the features selected by humans and DNNs on ImageNet. In general, humans selected less context and focused more on object parts: for animals, parts of their faces; for non-animals, parts that enable their usage, like the spade of a shovel (see Fig. 2 and SI Fig. 5. The DNN that was most aligned with humans, the DenseNet121, was still only 38% aligned with humans (Fig. 3).

Plotting the relationship between DNNs' top-1 accuracy on ImageNet with their human alignment revealed a striking trade-off: as the accuracy of DNNs has improved beyond DenseNet121, their

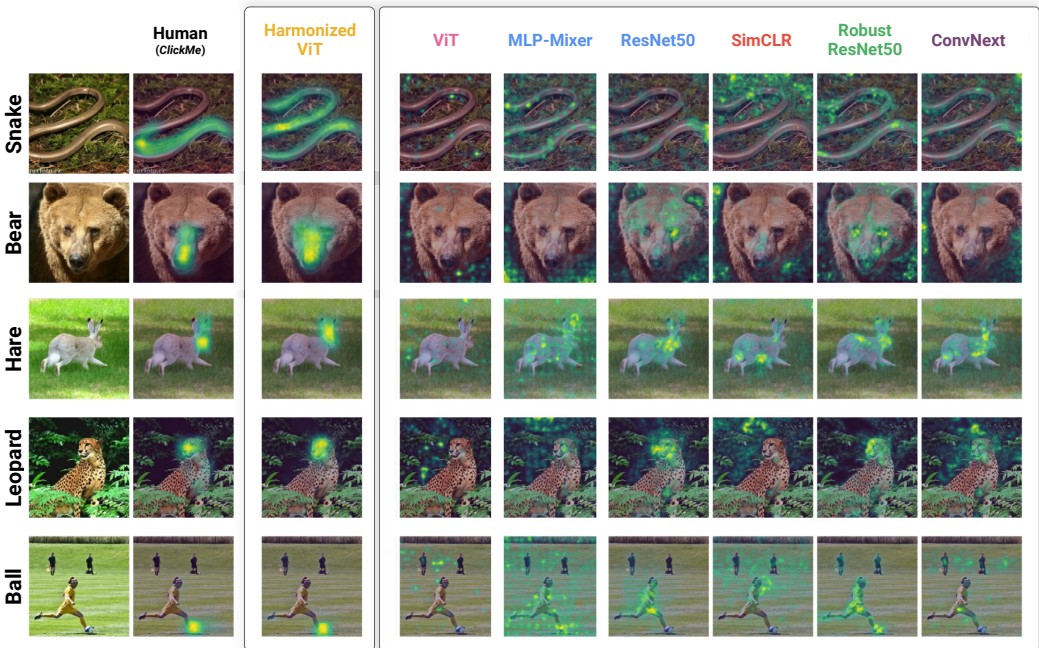

Figure 2: **Human and DNNs rely on different features to recognize objects.** In contrast, our neural harmonizer aligns DNN feature importance with humans. We smooth feature importance maps from humans (*ClickMe*) and DNNs with a Gaussian kernel for visualization.

alignment with humans has worsened (Fig. 3). For example, consider the ConvNext [1], which achieved the best top-1 accuracy in our experiments (85.8%), was only 22% aligned with humans – equivalent to the alignment of the BagNet33 [69] (63% top-1 accuracy). As an additional control, we computed the similarity between the average *ClickMe* map, which exhibits a center bias [85, 86] (SI Fig. 5), and each individual *ClickMe* map. This center-bias control was only outperformed by 42/84 CNNs we tested († in Fig. 3). Overall, we observe that human and DNN alignment has considerably worsened since the introduction of these two models.

**The neural harmonizer.** While scaling DNNs has immensely helped performance on popular benchmark tasks, there are still fundamental differences in the architectures of DNNs and the human visual system [37] which could part of the reason to blame for poor alignment. While introducing biological constraints into DNNs could help this problem, there is plenty of evidence that doing so would hurt benchmark performance and require bespoke development for every different architecture [87–89]. *Is it possible to align a DNN's visual strategies with humans without hurting its performance?*

Such a general-purpose method for aligning human and DNN visual strategies should satisfy the following criteria: (***i***) The method should work with any fully-differentiable network architecture. (***ii***) It should not present optimization issues that interfere with learning to solve a task, and the task-accuracy of a model trained with the method should not be worse than a model trained without the method. We created the neural harmonizer to satisfy these criteria.

Let us consider a supervised categorization problem with an input space, $\mathcal{X}$ an output space $\mathcal{Y} \subseteq \mathbb{R}^c$ and a predictor function $\boldsymbol{f}_\theta : \mathcal{X} \rightarrow \mathcal{Y}$ parameterized by $\theta$, which maps an input vector $\boldsymbol{x} \in \mathcal{X}$ to an output $\boldsymbol{f}_\theta(\boldsymbol{x})$. We denote $\boldsymbol{g} : \mathcal{F} \times \mathcal{X} \rightarrow \mathcal{X}$ an explanation functional that, given a predictor $\boldsymbol{f}_\theta \in \mathcal{F}$ and an input, returns a feature importance map $\boldsymbol{\phi} = \boldsymbol{g}(\boldsymbol{f}_\theta, \boldsymbol{x})$. Here, we focus on DNN saliency $\boldsymbol{g}(\boldsymbol{f}_\theta, \boldsymbol{x}) \triangleq \nabla_{\boldsymbol{x}} \boldsymbol{f}_\theta(\boldsymbol{x})$ as our method for computing feature importance in DNNs, but the method can in principle work with any differentiable network explanation method.

To satisfy criterion (***i***), the neural harmonizer introduces a differentiable loss that will enforce alignment across feature importance map scales from any neural network. Let $\mathcal{P}_i(.)$ be the function mapping a feature importance map $\boldsymbol{\phi}$ to it is representation in the $N$ levels of a Gaussian pyramid, with $i \in \{1, ..., N\}$. The function $\mathcal{P}_i(\boldsymbol{\phi})$ is computed by downsampling $\mathcal{P}_{i-1}(\boldsymbol{\phi})$ using a Gaussian

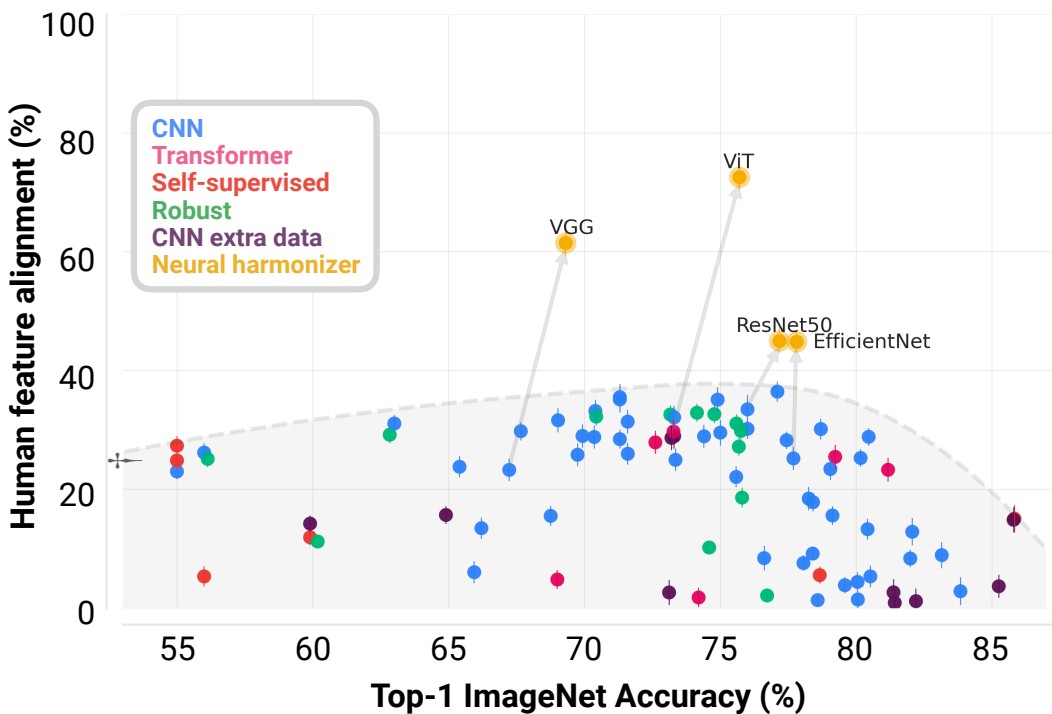

Figure 3: **The trade-off between DNN performance and alignment with human feature importance from *ClickMe* [20]**. Human feature alignment is the mean Spearman correlation between human and DNN feature importance maps, normalized by the average inter-rater alignment of humans. The shaded region denotes the pareto frontier of the trade-offs between ImageNet accuracy and human feature alignment for unharmonized models. Harmonized models (VGG16, ResNet50, ViT, and EfficientNetB0) are more accurate and aligned than versions of those models trained only for categorization. Error bars are bootstrapped standard deviations over feature alignment. Arrows show a shift in performance after training with the neural harmonizer. The feature alignment of an average of *ClickMe* maps with held-out maps is denoted by †.

kernel, with $\mathcal{P}_1(\phi) = \phi$. We then seek to minimize $\sum_i^N ||\mathcal{P}_i(\boldsymbol{g}(\boldsymbol{f}_\theta, \boldsymbol{x})) - \mathcal{P}_i(\phi)||^2$, which will align feature importance maps between humans and DNNs at every scale of the pyramid.

To satisfy criterion (*ii*), the neural harmonizer should work well with training routines designed for large-scale computer vision challenges like ImageNet. This means that the neural harmonizer loss must avoid optimization issues at scale. To do this, we need a way of comparing feature importance maps between humans and DNNs that is invariant to the norm of either map. We therefore standardize feature importance maps from humans and DNNs before comparing them, and only measure alignment on the most important areas of the image for each observer. Formally, let $\boldsymbol{z}(.)$ be a standardization function over feature importance maps that takes the mean and standard deviation computed for each map $\phi$ such that $\boldsymbol{z}(\phi)$ has 0 activation on average and unit standard deviation. To focus alignment on important regions, let $\boldsymbol{z}(\phi)^+$ denote the positive part of the standardized explanation $\boldsymbol{z}(\phi)$. Finally, we include a task loss, the familiar cross entropy objective, to yield the complete neural harmonization loss and train models that are at least as accurate as those trained without harmonization:

$$\mathcal{L}_{\text{Harmonization}} = \lambda_1 \sum_i^N ||(\boldsymbol{z} \circ \mathcal{P}_i \circ \boldsymbol{g}(\boldsymbol{f}_\theta, \boldsymbol{x}))^+ - (\boldsymbol{z} \circ \mathcal{P}_i(\phi))^+||_2 \tag{1}$$

$$+ \mathcal{L}_{CCE}(\boldsymbol{f}_\theta, \boldsymbol{x}, \boldsymbol{y}) + \lambda_2 \sum_i \theta_i^2 \tag{2}$$

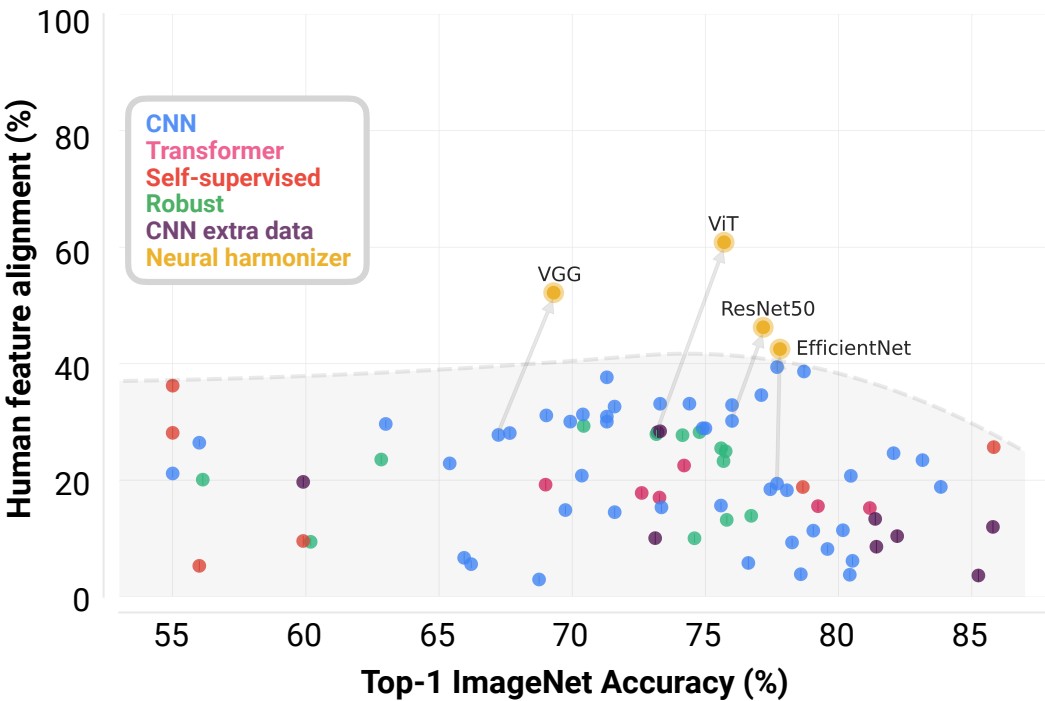

Figure 4: **The trade-off between DNN performance and alignment with human feature importance from *Clicktionary* [22]**. Human feature alignment is the mean Spearman correlation between human and DNN feature importance maps, normalized by the average inter-rater alignment of humans. The shaded region denotes the pareto frontier of the trade-offs between ImageNet accuracy and human feature alignment for unharmonized models. Harmonized models (VGG16, ResNet50, MobileNetV1, and EfficientNetB0) are more accurate and aligned than versions of those models trained only for categorization. Error bars are bootstrapped standard deviations over feature alignment. Arrows denote a shift in performance after training with the neural harmonizer.

**Training.** We trained four different DNNs with the neural harmonizer: VGG16, ViT, ResNet50, and EfficientNetB0. These models were selected because they are popular convolutional and transformer networks with open-source architectures that are straightforward to train and also sit near the boundary of the trade-off between DNN performance and alignment with humans. Models were trained using the neural harmonizer to optimize categorization performance on ImageNet and feature importance map alignment with human data from *ClickMe*. We trained models on all images in the ImageNet training set, but because *ClickMe* only contains human feature importance maps for a portion of those images, we computed the categorization loss but not the neural harmonizer loss for images without importance maps. Models were trained using 8 cores V4 TPUs on the Google Cloud Platform, and training lasted approximately one day. Models were trained with an augmented ResNet training recipe (built from `https://github.com/tensorflow/tpu/`). Models were optimized with SGD and momentum over batches of 512 images, a learning rate of 0.3, and label smoothing [90]. Images were augmented with random left-right flips and mixup [91]. The learning rate was adjusted over the course of training with a schedule that began with an initial warm-up period of 5 epochs and then decaying according to a cosine function over 90 epochs, with decay at step 30, 50 and 80. We validated that a ResNet50 and VGG16 trained with these hyperparameters and schedule using standard cross-entropy (but not the neural harmonizer) matched published performance.

**The neural harmonizer aligns human and DNN visual strategies.** We found that harmonized models broke the trade-off between ImageNet accuracy and model alignment with *ClickMe* human feature importance maps (Fig. 3). Harmonized models were significantly more aligned with feature importance maps and also performed better on ImageNet. The changes in *where* harmonized models find important features in images were dramatic: a harmonized ViT had feature importance maps that are far less reliant on context (Fig. 2) and approximately 150% more aligned with humans (Fig. 3;

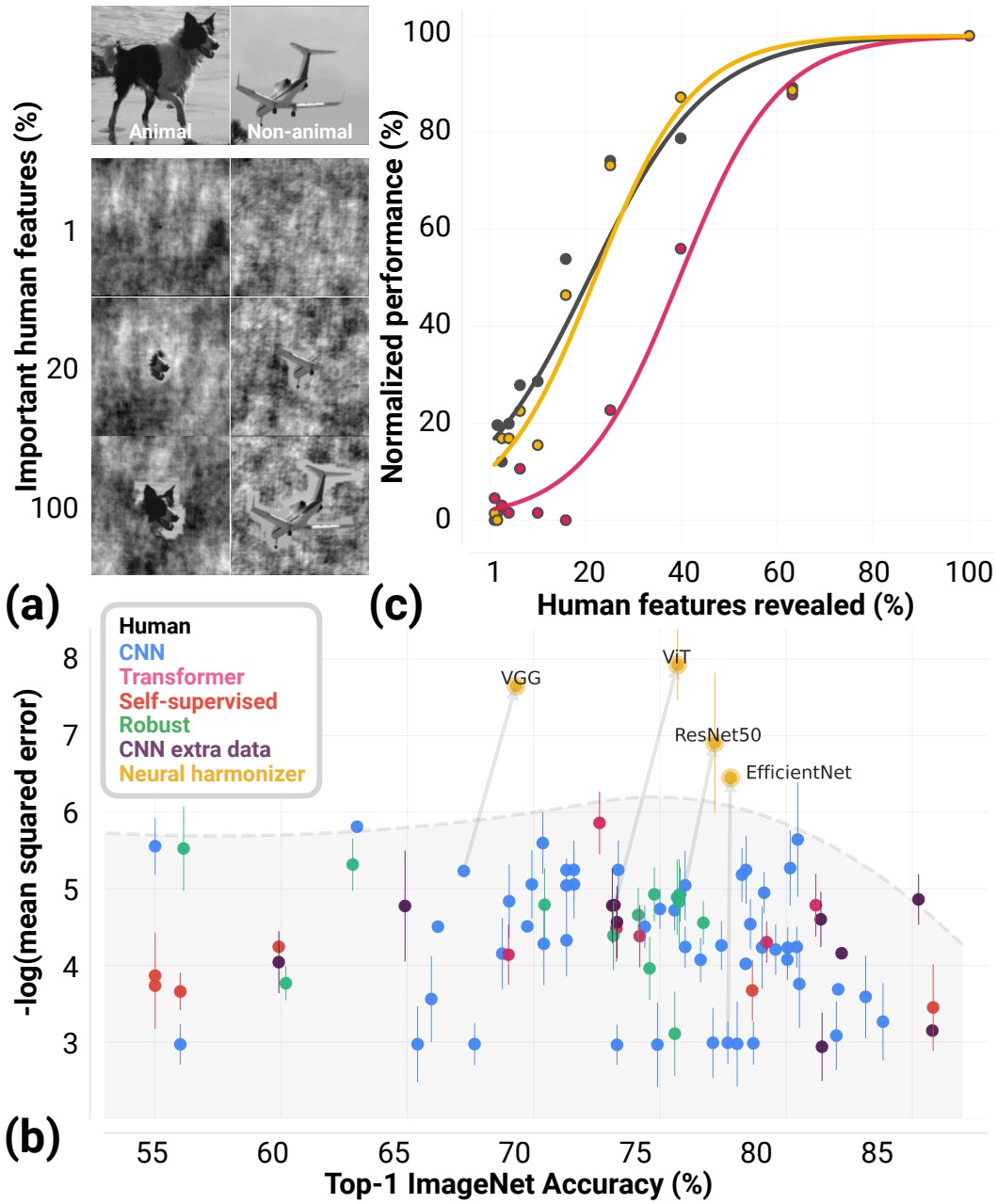

Figure 5: **Comparing *how* humans and DNNs use visual features during object recognition**. **(a)** Humans and DNNs categorized ImageNet validation images as animals or non-animals. The images revealed only a portion of the most important visual features according to the *Clicktionary* game [92]. **(b)** There was a trade-off between DNN top-1 accuracy on ImageNet and alignment with human visual decision making. The shaded region denotes the pareto frontier of the trade-off between ImageNet accuracy and human feature alignment for unharmonized models. Arrows denote a shift in performance after training with the neural harmonizer. Error bars are bootstrapped standard deviations over decision-making alignment. **(c)** A state-of-the-art DNN like the ViT learned a different strategy for integrating visual features into decisions than humans or a harmonized ViT.

ViT goes from 28.7% to 72.6% alignment after harmonization). The same model also performed 4% better in top-1 accuracy without any changes to its architecture. Similar improvements were found for

the harmonized VGG16 and ResNet50. While the EfficientNetB0 had only a minimal improvement in accuracy, it too exhibited a large boost in human feature alignment.

**Clicktionary.**    To test if the trade-off between DNN ImageNet accuracy and alignment with humans is a general phenomenon we next turned to *Clicktionary* [22]. Indeed, we observed a similar trade-off on this dataset as we found for *ClickMe*: alignment with human feature importance from *Clicktionary* has worsened as DNN accuracy has improved on ImageNet (Fig. 4). As with *ClickMe*, harmonized DNNs shift the accuracy-alignment trade-off on this dataset.

### 4.2   *How* do humans and DNNs integrate diagnostic object features into decisions?

The trade-off we discovered between DNN accuracy on ImageNet and alignment with human visual feature importance suggests that the two use different visual strategies for object classification. However, there is potential for an even deeper problem. Even if two observers deem the same regions of an image as important for recognizing it, there is no guarantee that they use the selected features in the same way to render their decisions. We posit that if two observers have aligned visual strategies, the will agree on both *where* important features are in an image and *how* they use those features for decisions.

We developed a psychophysics experiment to measure how different humans use features in ImageNet images to recognize objects. Participants viewed versions of these images where only a proportion of the features that were deemed most important in the *Clicktionary* game were visible (Fig. 5a). Participants had to accurately detect whether or not the image contained an animal within 550ms, which forced them to rely on feedforward processing as much as possible [33]. Each of the 200 images we used were shown to a single participant only once. We accumulated responses from all participants to construct decision curves that showed how accurately the average human converted any given proportion of image features into an object decision. We performed the same experiment on DNNs as we did on humans, recording animal vs. non-animal decisions according to whether or not the most probable category in the model's 1000-category output was an animal. Because the experiment was speeded, humans did not achieve perfect accuracy. Thus, we normalized performance for humans and DNNs to compare the rate at which each integrated features into accurate decisions.

We discovered a similar trade-off between ImageNet accuracy and alignment with human visual decision making in this experiment as we did in *ClickMe* and *Clicktionary* (Fig. 5b). Indeed, the model that was most aligned with human decision-making – the BagNet33 [69] – only achieved 63.0% accuracy on ImageNet. Surprisingly, harmonized models broke this trend, particularly the harmonized ViT (Fig. 5b, top-right), despite no explicit constraints in that procedure which forced consistent decision-making with humans. In contrast, an unharmonized ViT integrates visual information into accurate decisions less efficiently than humans or harmonized models (Fig. 5c).

## 5   Conclusion

Models that reliably categorize objects like humans do would shift the paradigms of the cognitive sciences and artificial intelligence. But despite continuous progress over the past decade on the ImageNet benchmark, DNNs are becoming *worse* models of human vision. Our finding resembles a growing number of concurrent works showing similar trade-offs between DNN performance and predictions of human perception on different tasks [93, 94]. Our solution to this problem, the neural harmonizer, can be applied to any DNN to align their visual strategies with humans and even improve performance.

We observed the greatest benefit of harmonization on the visual transformer, the ViT. This finding is particularly surprising given that transformers eschew the locality bias of convolutional neural networks that has helped them become the new standard for modeling human vision and cognition [37]. Thus, we suspect that the neural harmonizer is especially well-suited for large-scale training on low-inductive bias models, like transformers. We also hypothesize that the improvements in human alignment provided by the neural harmonizer will yield a variety of downstream benefits for a model like the ViT, including better predictions of perceptual similarity, stimulus-evoked neural responses, and even performance on visual reasoning tasks. We leave these analyses for future work.

The field of computer vision today is following Sutton's prescient lesson: benchmark tasks can be scaling architectural capacity and the size of training data. However, as we have demonstrated here, these scaling laws are exchanging performance for alignment with human perception. We encourage the field to re-analyze the costs and benefits of this exchange, particularly in light of the growing concerns about DNNs leveraging shortcuts and dataset biases to achieve high performance [19]. Alignment with human vision need not be exchanged with performance if DNNs are harmonized. Our codebase (`https://serre-lab.github.io/Harmonization/`) can be used to incorporate the neural harmonizer into any DNN created and measure its alignment with humans on the datasets we describe in this paper.

**Limitations.** One possible explanation for the misalignment between DNNs and humans that we observe is that recent DNNs have achieved superhuman accuracy on ImageNet. Superhuman DNNs have been described in biomedical applications [95, 96] where there is definitive biological ground-truth labels, but ImageNet labels are noisy, making it unclear if such an achievement is laudable (`https://labelerrors.com/`). Thus, an equally likely explanation is that the continued improvements of DNNs at least partially reflect their exploitation of shortcuts in ImageNet [19].

The scope of our work is also limited in that it focuses on object recognition in ImageNet. It is possible that models trained on other tasks, such as segmentation, may be more aligned with humans.

Finally, our modeling efforts were hamstrung for the largest-scale models in existence. Our work does not answer how much harmonization would help a model like CLIP because of the massive investment needed to train it. The neural harmonizer can be applied to CLIP but it is possible that more *ClickMe* human feature importance maps are needed for successful harmonization.

**Broader impacts.** A persistent issue in the field of artificial intelligence is the tendency of models to exploit dataset biases. A central theme of our work is that there are facets of human perception that are not captured by DNNs, particularly those which follow the scaling laws which have been so embraced by industry leaders. Forcing DNNs to rely on similar visual strategies as humans could represent a scalable path forward to correcting the insidious biases which have assailed under-constrained models of artificial intelligence.

## Acknowledgments and Disclosure of Funding

This work was supported by ONR (N00014-19-1-2029), NSF (IIS-1912280 and EAR-1925481), DARPA (D19AC00015), NIH/NINDS (R21 NS 112743), and the ANR-3IA Artificial and Natural Intelligence Toulouse Institute (ANR-19-PI3A-0004). Additional support provided by the Carney Institute for Brain Science and the Center for Computation and Visualization (CCV). We acknowledge the Cloud TPU hardware resources that Google made available via the TensorFlow Research Cloud (TFRC) program as well as computing hardware supported by NIH Office of the Director grant S10OD025181.

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
