# Supplementary Information:
# Aligning human and machine vision

**Thomas Fel**[*,1,2], **Ivan Felipe**[*,1], **Drew Linsley**[*,1,3], **Thomas Serre**[1,2,3]
{thomas_fel,ivan_felipe_rodriguez,drew_linsley}@brown.edu

## 1  Psychophyics

The psychophysics experiments of §4.2 were implemented with the psiTurk framework [1] and custom javascript functions. Each trial sequence was converted to a HTML5-compatible video for the fastest reliable presentation time possible in a web browser. Videos were cached before each trial to optimize reliability of experiment timing within the web browser. A photo-diode verified the reliability of stimulus timing in our experiment was consistently accurate within $\sim 10\mathrm{ms}$ across different operating system, web browser, and display type configurations.

**Participants:**   We recruited 199 participants from Amazon Mechanical Turk (mturk.com) for the experiments. Participants were based in the United States, used either the Firefox or Chrome browser on a non-mobile device, and had a minimal average approval rating of 95% on past Mechanical Turk tasks.

**Stimuli:**   Experiment images were taken from the *Clicktionary* dataset [2]. Images were sampled from 5 target and 5 distractor categories: border collie, sorrel (horse), great white shark, bald eagle, and panther; trailer truck, sports car, speedboat, airliner, and school bus. Images were presented to human participants (and DNNs) either intact or with a perceptual phase scrambled mask that exposed a proportion of their most important visual features, as described in the main text. Images were cast to greyscale to control for trivial color-based cues for classification and blend the scrambled mask background into the foreground. Responses to intact images were used to normalize the performance of each observer on masked images relative to their maximum performance on these images.

Image masks were created for each image to reveal only a proportion of the most important visual features. For each image, we created masks that revealed between 1% and 100% (at log-scale spaced intervals) of the object pixels in the corresponding image's *Clicktionary* feature importance map. We generated these masks in two steps. First, we computed a phase-scrambled version of the image [3,4]. Next, we used a novel "stochastic flood-fill" algorithm to reveal a contiguous region of the most important visual features in the image according to humans. Our flood-fill algorithm was seeded on the pixel deemed most important by humans in the image, then grew outwards anisotropically and biased towards pixels with higher feature importance scores (Figure 1). The revealed region was always centered on the image. Each participant saw every category exemplar only once, with its amount of image revelation randomly selected from all possible configurations.

After providing online consent, participants were instructed to complete a rapid visual categorization task in which they had to classify stimuli revealing a portion of the most diagnostic object features (Fig. 3). Each experimental trial began with a cross for participants to fixate for a variable time (1,100–1,600ms), then a stimulus for 400ms, then another cross and additional time for participants to render a decision. Participants were instructed to provide a decision after the first fixation cross,

---

[*]These authors contributed equally.

[1]Department of Cognitive, Linguistic, & Psychological Sciences, Brown University, Providence, RI

[2]Artificial and Natural Intelligence Toulouse Institute (ANITI), Toulouse, France

[3]Carney Institute for Brain Science, Brown University, Providence, RI

36th Conference on Neural Information Processing Systems (NeurIPS 2022).

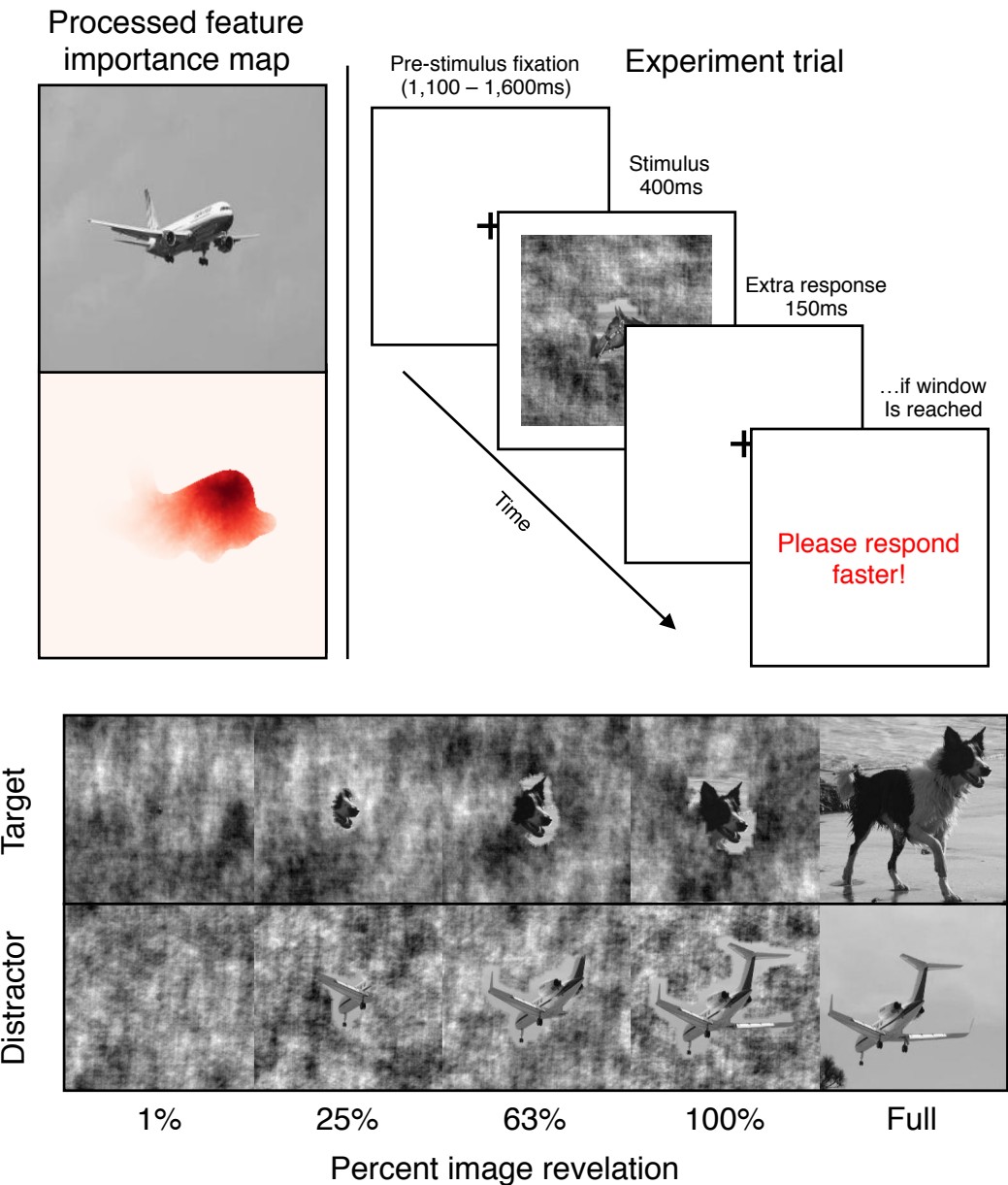

Figure 1: **Overview of the psychophysics paradigm.** Participants performed a rapid animals vs. vehicles categorization paradigm (top). Stimuli were created using feature importance maps derived from humans or DNNs via a "stochastic flood-fill" algorithm that revealed image regions of different sizes centered on important features. Sample stimuli are shown (bottom) for different percentages of image revelation. Note that 100% revelation corresponds to all non-zero pixels in a feature importance map.

but that they only had 650ms to answer. If they were too slow to respond they were told to respond faster and the trial was discarded.

## 2   Harmonization loss

The neural harmonizer loss Fig. 2 uses several components crucial to its performance: a pyramidal representation of decision explanation maps and normalizing those maps.

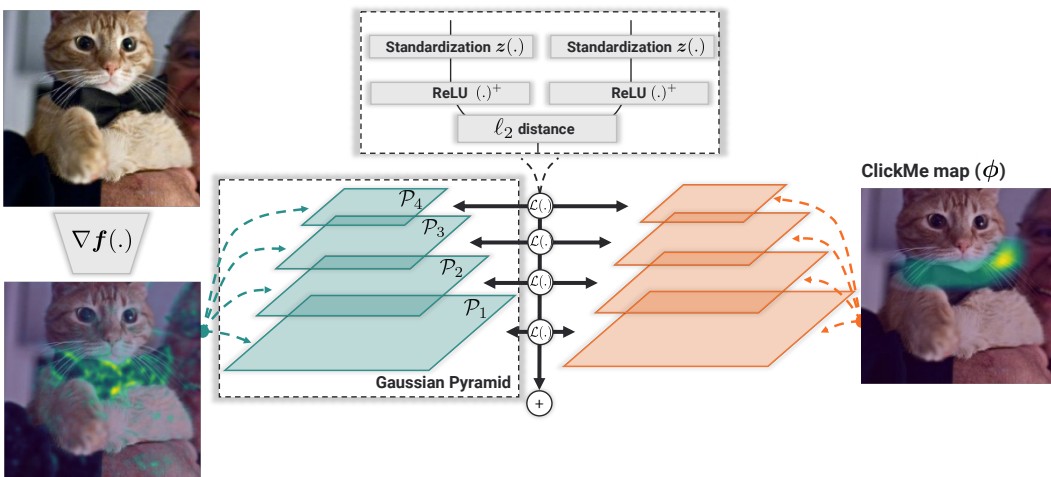

Figure 2: **Computing the neural harmonizer loss..**

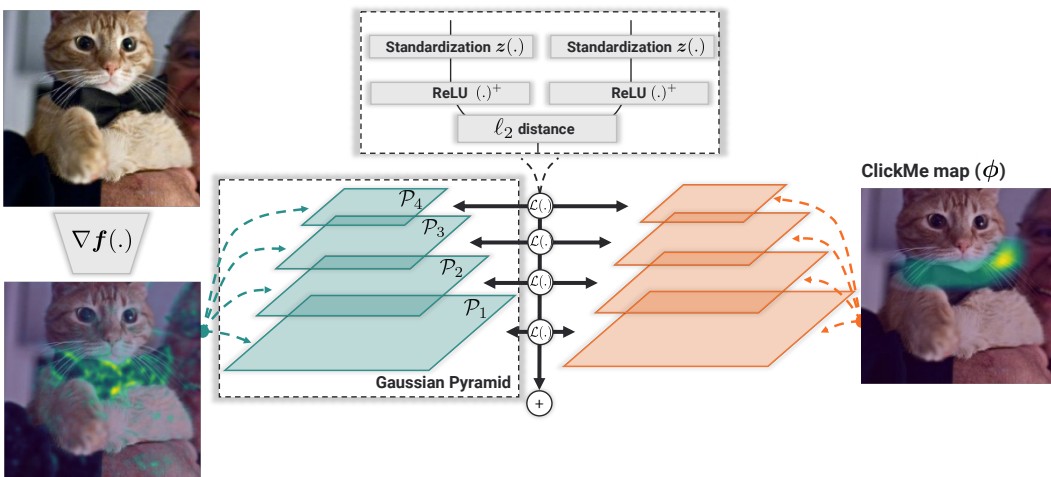

Figure 3: **Psychophysics experiment instructions.**

When computing the difference between model explanations for an image and the human feature importance map for that image, we rely on a pyramid representation of each to compute these differences Fig. 2). This pyramid allows for a model to align its feature representations with humans at multiple scales and corrects for an important problem in datasets like *ClickMe*: the human data is an approximation and not precise at the pixel level. This lack of precision can present optimization issues, and computing a pyramid representation alleviates those issues because it allows a model to learn to focus on regions that are important for humans without pixel-level precision.

Standardization tackles a similar problem: because of the imprecision of human data, we choose to focus harmonization on only the most important areas selected by humans in *ClickMe*. By standardizing then rectifying before comparing human and model explanations, we reduce noise in the harmonization procedure.

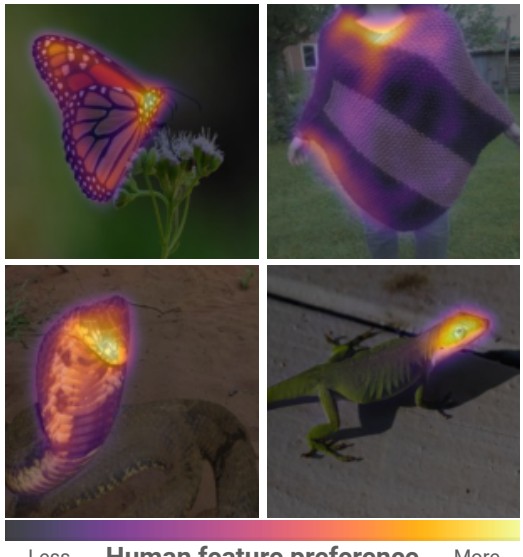

Figure 4: **Example *ClickMe* feature importance maps on ImageNet images.**

## 3 Additional Results

### 3.1 *ClickMe*

The *ClickMe* game by [5] was used to identify category diagnostic features in ImageNet images. These feature importance maps largely focus on object regions rather than context, and in contrast to segmentation maps select features on the "front" or "face" of objects (Fig. 4).

As discussed in the main text, we found a trade-off between DNN top-1 ImageNet accuracy and the alignment of their feature importance maps with humans importance maps from *ClickMe*. This trade-off persists across multiple scales of feature importance maps, including 4× (Fig. 6) and 16× (Fig. 7) sub-sampled maps, meaning that simple smoothing is not sufficient to fix the trade-off.

### 3.2 *ViT attention*

While in the main text we investigate alignment between humans and models using gradient feature importance visualizations, the attention maps in transformer models like the ViT provide another avenue for investigation. To understand whether or not attention maps from ViT are more aligned with humans than their gradient-based decision explanation maps, we computed attention rollouts for harmonized and unharmonized ViTs [6]. We found that both versions of the ViT had similar correlations between their attention rollouts and human *ClickMe* maps: 0.38 for the harmonized ViT and 0.393 for the unharmonized model. This surprising result suggests that the harmonizer affects the process by which ViTs integrate visual information into their decisions rather than how they allocate attention. Through manipulating ViT decision making processes, the harmonizer can induce the large changes in gradient-based visualizations and psychophysics that we describe in the main text.

### 3.3 *Correlations between measurements of human visual strategies*

Our results rely on three independent datasets measuring different features of human visual strategies: *ClickMe*, *Clicktionary*, and the psychophysics experiments we introduce in this manuscript. The fact that all three evoke similar trade-offs between top-1 accuracy and human alignment is a surprising result that deserves further attention. We investigated these trade-offs by measuring the correlation between human alignment on each dataset, with and without models trained with the neural harmonizer. We found that correlations between datasets were lower across the board when neural harmonizer models were not included. The association between model alignments with *Clicktionary* versus psychophysics results were not significant ($\rho = 0.21$, n.s.; Fig. 9), but the associations between

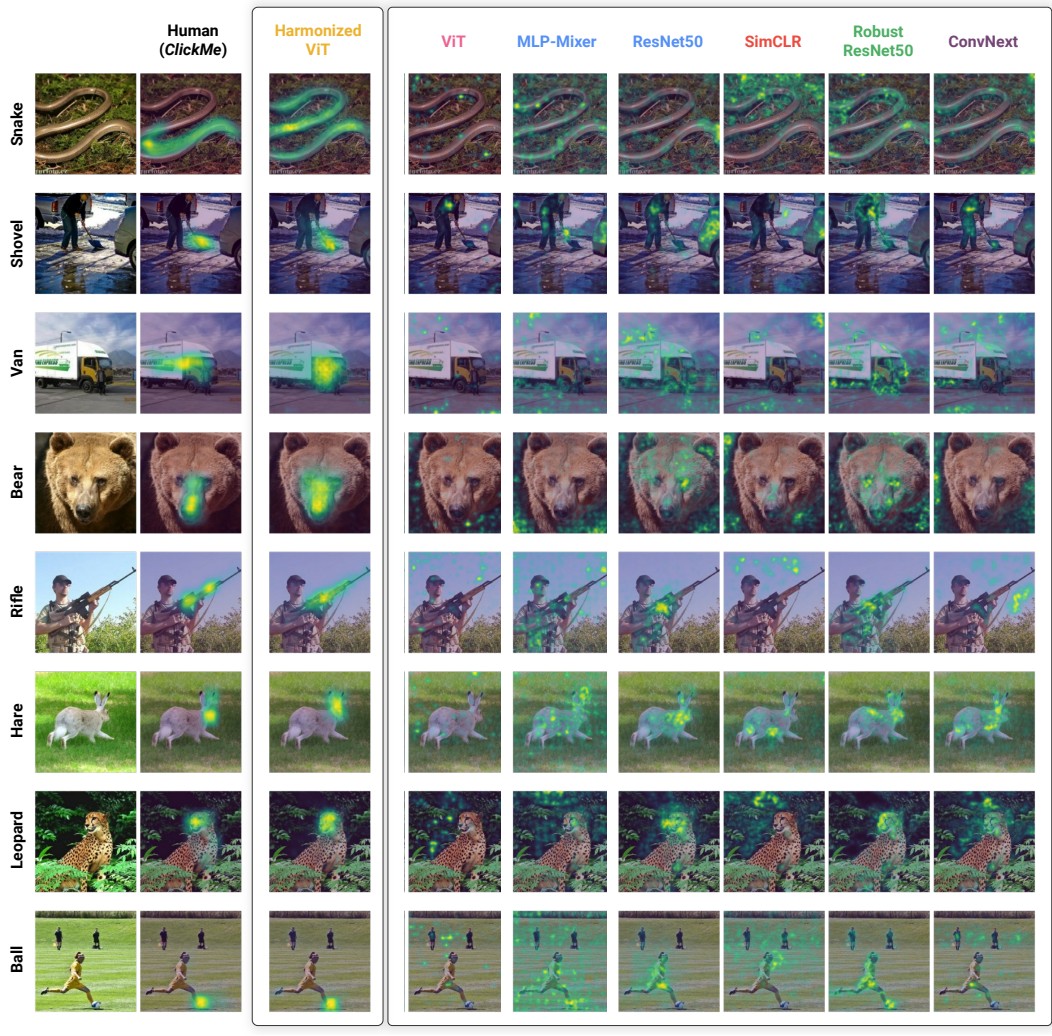

Figure 5: **Feature importance maps of humans, harmonized, and unharmonized models on ImageNet**.

model alignments with *ClickMe* versus psychophysics ($\rho = 0.51, p < 0.001$; Fig. 8) and *ClickMe* versus *Clicktionary* ($\rho = 0.77, p < 0.001$; Fig. 10) were both significant. Each correlation improved when the neural harmonizer models were included in the calculation. This finding indicates that the neural harmonizer successfully aligned visual strategies between humans and DNNs, and was not merely benefiting from either *where* humans versus DNNs considered important visual features to be or *how* humans versus DNNs incorporated those features into their decisions.

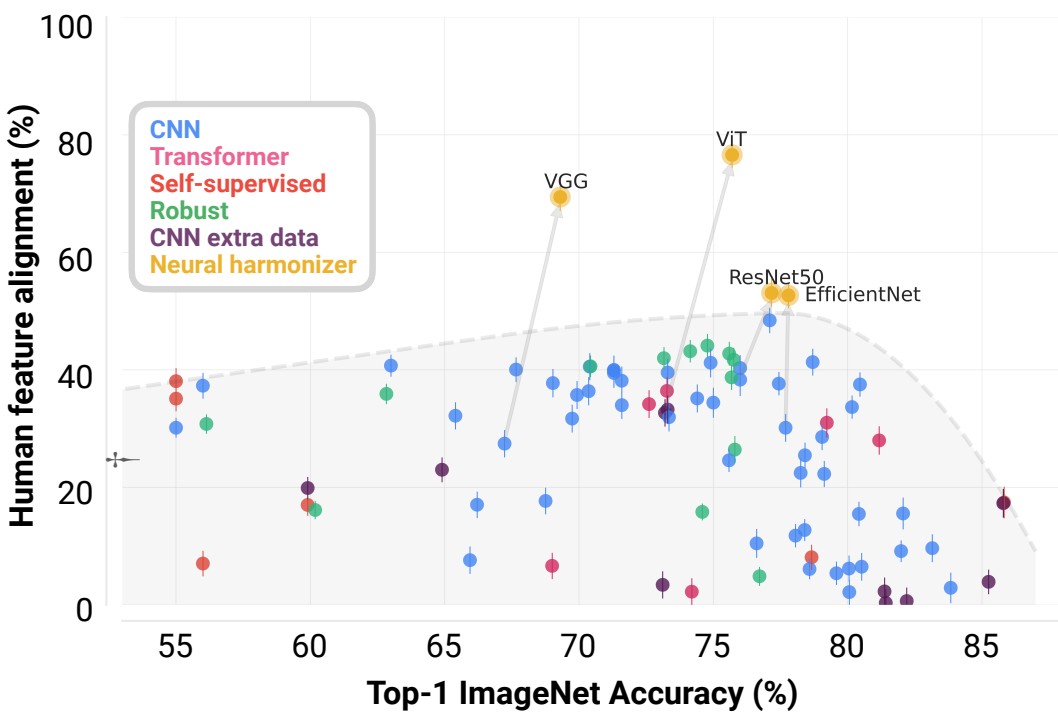

Figure 6: **The neural harmonizer's effect is robust across image scales.** Here, we show that the trade-off between ImageNet accuracy and alignment with humans holds across downsizing by a factor of *4*. The Neural harmonizer once again yields the model with the best alignment with humans. Grey-shaded area captures the trade-off between accuracy and alignment in standard DNNs. Error bars are bootstrapped standard deviations over feature alignment.

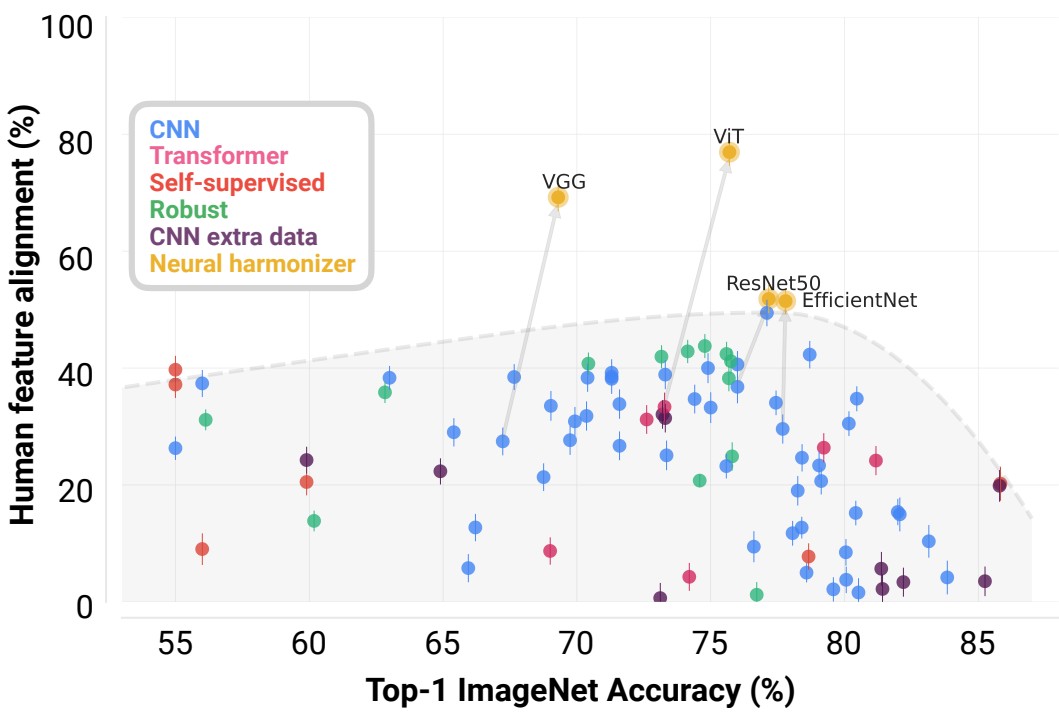

Figure 7: **The neural harmonizer's effect is robust across image scales.** Here, we show that the trade-off between ImageNet accuracy and alignment with humans holds across downsizing by a factor of *16*. The Neural harmonizer once again yields the model with the best alignment with humans. Grey-shaded area captures the trade-off between accuracy and alignment in standard DNNs. Error bars are bootstrapped standard deviations over feature alignment.

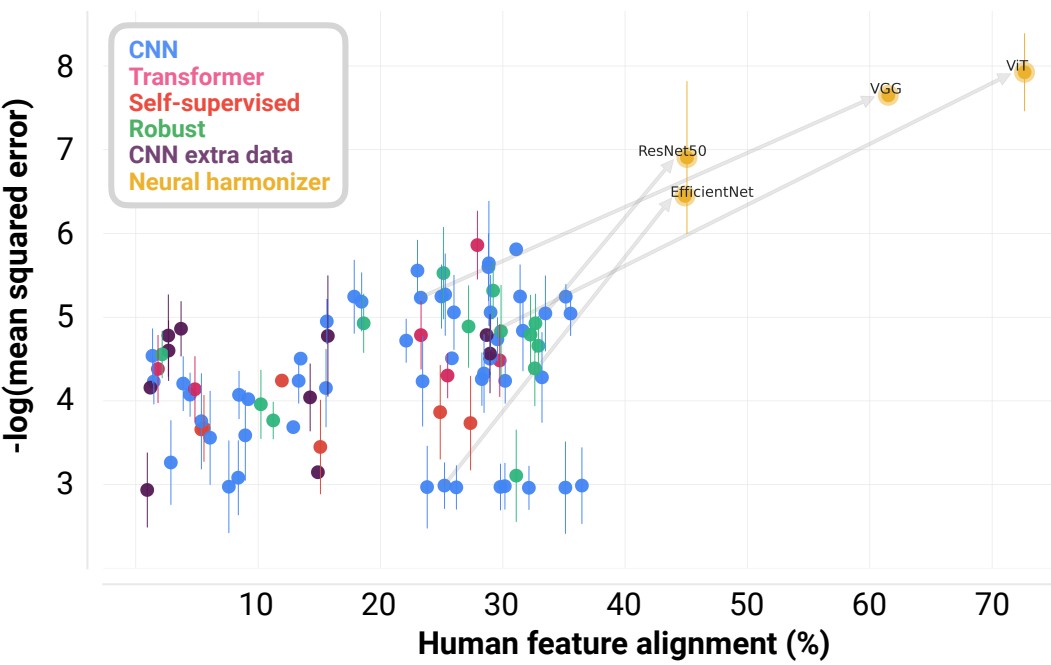

Figure 8: **The association between *ClickMe* alignment versus psychophysics alignment.** These scores are significantly correlated, $\rho = 0.68, p < 0.001$. Error bars are bootstrapped standard deviations over feature alignment.

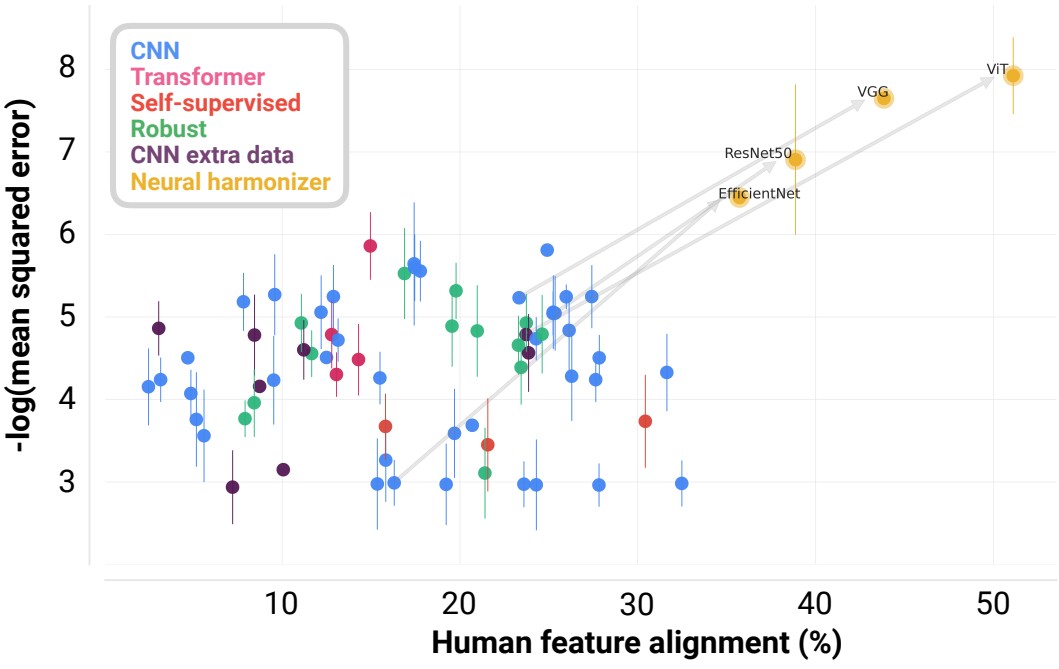

Figure 9: **The association between *Clicktionary* alignment versus psychophysics alignment.** These scores are significantly correlated, $\rho = 0.53, p < 0.001$.

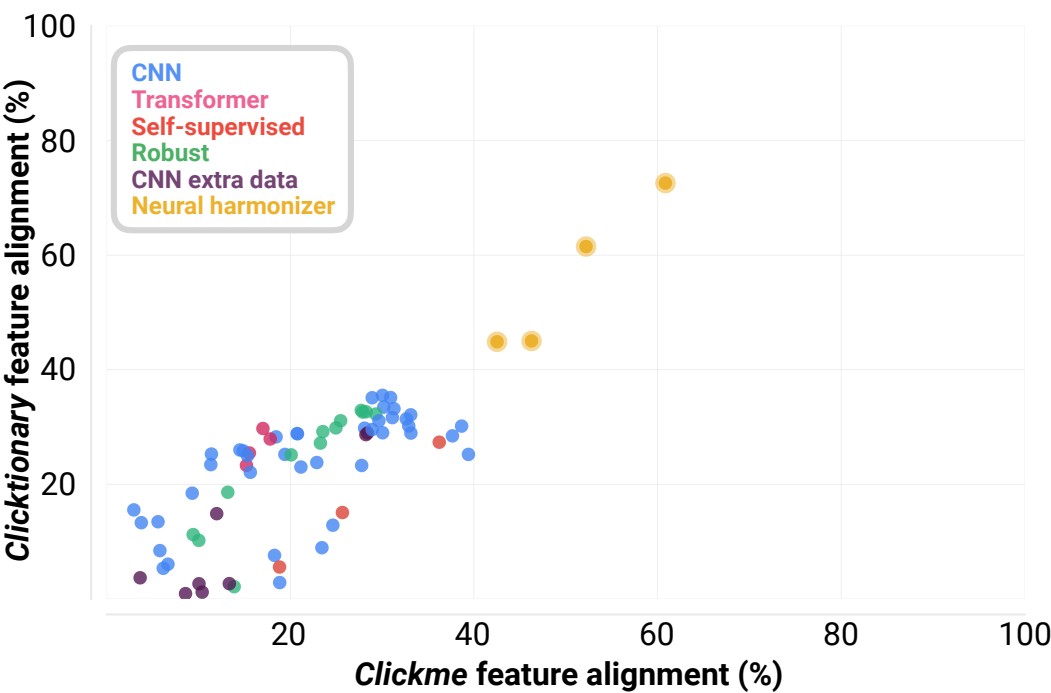

Figure 10: **The association between *ClickMe* alignment versus *Clicktionary* alignment.** These scores are significantly correlated, $\rho = 0.85, p < 0.001$. Error bars are bootstrapped standard deviations over feature alignment.

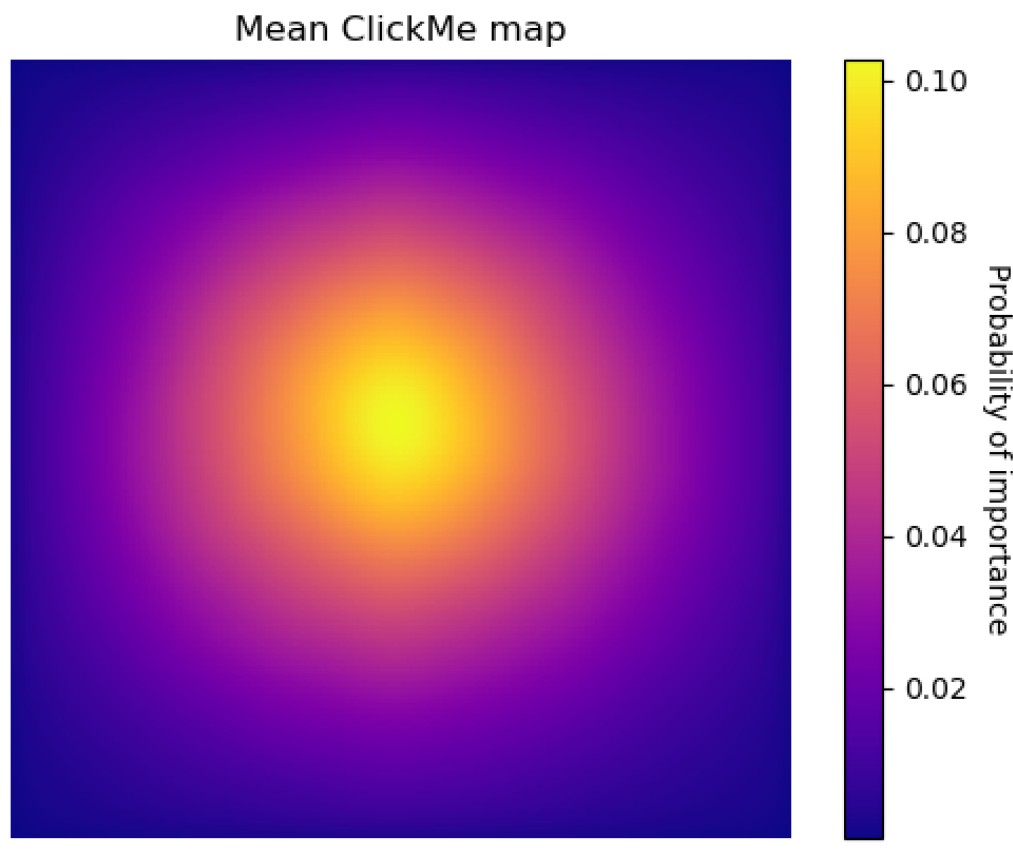

Figure 11: **The mean of *ClickMe* feature importance maps exhibits a center bias, likely due to the positioning of objects in ImageNet images rather than a purely spatial bias of human participants (compare to individual maps shown in Fig. 4)**.