# OpenReview forum: "Harmonizing the object recognition strategies of deep neural networks with humans"
_NeurIPS.cc/2022/Conference — NeurIPS 2022 Accept_

### Official Review · Reviewer_cEc7 · 2022-07-07

**Rating:** 7
**Confidence:** 4
**Soundness:** 4 excellent
**Presentation:** 4 excellent
**Contribution:** 2 fair

**Summary:**

Summary: This paper focusses on evaluating the alignment between visual strategies of DNNs and human perception. Specifically, the ask how this alignment evolves as architectures become more complex. For a fair comparison the paper focusses on only the rapid, feed-forward component of human perception, and suggests evaluating alignment on visual strategies as defined by a two step process---identifying regions with important features and how these regions are used to make a classification decision. For the first part, the paper evaluates alignment between DNNs and humans on three datasets. These include ClickMe and Clicktionary datasets with ground truth annotations for human visual importance, and a third dataset collected by the authors using a psychophysics experiment involving rapid classification of images. The paper shows that identifying important features peaks as architectures become better at classification, but then start dipping as they get more complex. To address this, the neural harmonizer methodology is proposed. Finally, the paper tests how these features are used by exposing different models to an increasing percentage of human-identified important regions, showing similar trends as the previous experiments.

**Questions:**

1. On the role of scene context: Human annotations highlighted lesser context areas (Page 5), but it is widely known that human visual recognition is strongly dependent on scene context. Does this suggest that what humans "think" of as visually important, may not essentially be the most important regions? Also may be suggested by work on minimal images, as it is counter-intuitive to humans that a few pixels can cause such large drops in accuracy. Would be good to see this discussed briefly in the limitations.


2.  A broader question is that the evaluation of alignment necessarily requires a gold standard of knowing what humans rely on the first place. How confident are we in the current measured ClickMe/Clicktionary style data? Also, how do the psychophysics data collected here improve upon these? A short discussion on this would be helpful.

**Limitations:**

Yes.

**Strengths And Weaknesses:**

Strengths:

1. Extremely interesting problem. While accuracy on standardized datasets is a good diagnostic for progress, measuring if these improvements result in better alignment between biological and artificial intelligence is a fundamental goal for a multitude of reasons. This has also been very well motivated in the paper.

2. The writing is clear and easy to follow. The work is placed well into the context of existing works, and is clear in elaborating how this work builds on top of this literature.

3. Above all, the experimental design is very solid. Controls are sufficient, and claims are well supported with either experimental results or through existing literature. Psychophysics experiments are well designed, and good care has been taken to ensure as fair a comparison between humans and DNNs as possible.

4. The neural harmonizer module is interesting as it increases alignment significantly.


Weaknesses:

1. The code and data are propriety and will NOT be released. The findings of this paper are interesting, but the most important contribution here seems to be an interesting benchmark for future researchers to focus on. Since the code and the data will not be released, it severely diminishes the utility for the research community.

Minor Weakness:

Some relevant recent literature is missing, and would be good to add to the paper:

A. On minimal images:

- Srivastava, S., Ben-Yosef, G. and Boix, X., 2019. Minimal images in deep neural networks: Fragile object recognition in natural images. arXiv preprint arXiv:1902.03227.

B. On alignment in humans and machines in existing works:

- Bomatter, P., Zhang, M., Karev, D., Madan, S., Tseng, C. and Kreiman, G., 2021. When pigs fly: Contextual reasoning in synthetic and natural scenes. In Proceedings of the IEEE/CVF International Conference on Computer Vision (pp. 255-264).

---

> ### Author Response · Authors · 2022-08-02
> **Response**
>
> > The code and data are propriety and will NOT be released.
>
> We included a link to the code in the discussion of the original manuscript. We have moved it to the abstract for visibility. Here is the link: https://anonymous.4open.science/r/Meta-predictor-3791/
>
> > Some relevant recent literature is missing
>
> We have added these references to the related work section, both of which are fantastic and we apologize for missing them the first time.
>
> Note that the “When pigs fly…” paper is about adjusting model accuracy to match humans when contextual information is available. We are not using human accuracy as the barometer in our work, but rather, human visual strategies. Our goal is to allow the field to scale model capacity as it is currently doing, to generate models with super-human accuracy that can nonetheless exploit predictably human-like visual strategies.
>
> >On the role of scene context: Human annotations highlighted lesser context areas (Page 5), but it is widely known that human visual recognition is strongly dependent on scene context. Does this suggest that what humans "think" of as visually important, may not essentially be the most important regions? Also may be suggested by work on minimal images, as it is counter-intuitive to humans that a few pixels can cause such large drops in accuracy. Would be good to see this discussed briefly in the limitations.
>
> Humans are indeed sensitive to context when recognizing objects, but context is of course not necessary for object recognition [1]. The *ClickMe* maps we used in this experiment contain sufficient features for accurately recognizing objects in a rapid vision paradigm that is similar to what we used in Figure 5 (see Figure 2 in [1] and our discussion with **Xrpg** on a similar question). Regarding minimal image patches, in the *Clicktionary* work they compared their feature importance maps to the minimally reducible configurations (MIRCs) of the original Ullman 2016 paper and found a strong correlation between the two. Thus, it is likely that the unifying feature of MIRCs, *ClickMe*, *Clicktionary*, etc. feature importance datasets is that they capture task-based saliency: the features that human observers rely on to recognize object images. The key difference between each of these efforts then is the ability to scale and the corresponding amount of noise that such scaling introduces. However, as the authors of *ClickMe* found, there is a strong correlation between their maps and *Clicktionary* despite capturing three orders-of-magnitude more data (and four orders-of-magnitude more data than the original MIRC study).
>
> > A broader question is that the evaluation of alignment necessarily requires a gold standard of knowing what humans rely on in the first place. How confident are we in the current measured ClickMe/Clicktionary style data? Also, how do the psychophysics data collected here improve upon these? A short discussion on this would be helpful.
>
> As mentioned above, the authors of the *ClickMe* dataset noted that their maps have high intersubject reliability and validated their utility for object classification in psychophysics experiments. They found that *ClickMe* maps contain sufficient information for object recognition, and significantly more diagnostic information than other large saliency datasets like SALICON contain.
>
> Our psychophysics experiments rely on feature importance maps from *Clicktionary* rather than *ClickMe*, as the method used to gather the former yields higher quality maps than the later. Our experiments test via forward inference whether a proportion of visual features in *Clicktionary* maps are sufficient for recognizing an object, which is one way of answering your question of what features humans rely on for recognition. Indeed, we found that humans achieve significantly above-chance accuracy when at least 10% of the important features in *Clicktionary* maps are visible. This result is similar to what is reported in Figure 2 in the *ClickMe* paper.
>
> To summarize, both the *ClickMe* dataset we used to harmonize DNNs and the *Clicktionary* dataset we used for our psychophysics depict visual features that are sufficient for object recognition.
>
> [1] Davenport J & Potter M. 2004. Scene consistency in object and background perception. Psychological science.

---

### Official Review · Reviewer_xZSj · 2022-07-07

**Rating:** 3
**Confidence:** 4
**Soundness:** 3 good
**Presentation:** 3 good
**Contribution:** 1 poor

**Summary:**

The authors conduct an empirical evaluation of the alignment of human and artificial neural network (ANN) saliency maps, and propose a method for improving it. The authors use saliency maps collected from ImageNet by Linsley et al. and compare them to those estimated from a variety of ANN's. In a separate experiment, the authors obscure images to varying degrees according to the human saliency maps, and measure the resulting degradation in classification performance in humans and ANN's, again comparing the two. Finally, the authors propose a method for aligning the saliency maps of humans and ANN's by jointly training the network for the original classification objective, and a measure of alignment with human saliency. The resulting models' saliency maps are indeed better aligned with human ones.

**Questions:**

As discussed above, the proposed neural harmonizer is of limited scientific interest, as it simply aligns training and testing for this particular metric (alignment with human saliency maps).

Assuming this were removed from the paper, could the authors instead re-analyze their data to study which training procedure, architecture, or data enables a model to reproduce human saliency maps without resorting to training on this data? For example

- Given the same model architecture and training data, which training procedure (supervised, self-supervised, etc) leads to the best alignment with human saliency?

- Given the same training data and objective (e.g. supervised or self-supervised), which model architecture (ResNet, ViT, etc) leads to the best alignment with human saliency?

- Given the same architecture and training objective, which training data (ImageNet, ImageNet21k, etc) leads to the best alignment with human saliency?

**Limitations:**

There are no ethical or societal concerns associated with this work.

**Strengths And Weaknesses:**

The authors propose a sound evaluation of the alignment between human and ANN saliency maps. In particular, the experiment they propose which incrementally reveals images according to human saliency maps appears to be a good way of comparing the features used by humans and ANN's in an apples-to-apples manner. Whether this experiments reveals "how features are used" rather than "where these features are" is less clear: it instead appears to be a better controlled version of the original comparison between saliency maps.

Secondly, the authors make good use of this evaluation to show that improving ImageNet classification accuracy doesn't lead to better alignment with human saliency maps, underscoring the limitations of ImageNet as a benchmark. It would be interesting to see if other computer vision benchmarks, such as object detection or semantic segmentation are more predictive of human saliency.

Finally, it is encouraging to see that co-training the network to reproduce saliency maps allows it to do so, without any cost to the ImageNet classification accuracy (or even with a slight improvement to it), however it is unclear what the scientific value of this result is. Aligning the training and testing objective usually improves test performance, hence improving the alignment with human saliency is hardly surprising. The fact that it improves ImageNet classification is more surprising, but is at odds with the first finding that models which improve classification accuracy do not improve the alignment with human saliency.

A much more interesting point would be to investigate what training procedure, architecture, or data enables a model to reproduce human saliency **without resorting to training on this data**. This would indicate which of the above is most closely matches the conditions leading to the human visual system's choice of features for performing these tasks.

---

> ### Author Response · Authors · 2022-08-02
> **Response**
>
> (We initially mistakenly pasted our responses to reviewer cZRY here. Sorry!)
>
> >The authors propose a sound evaluation of the alignment between human and ANN saliency maps. In particular, the experiment they propose which incrementally reveals images according to human saliency maps appears to be a good way of comparing the features used by humans and ANN's in an apples-to-apples manner. Whether this experiments reveals "how features are used" rather than "where these features are" is less clear: it instead appears to be a better controlled version of the original comparison between saliency maps.
>
> Sorry for the confusion. In the psychophysics experiment (results in Figure 5 and method diagram in SI Figure 1 included in the original submission), we asked humans and DNNs to recognize object images when only a proportion of their most important features were visible. The features used were selected and ranked according to **human** feature importance maps collected in the *Clicktionary* game, then were centered in each image over a phase-scrambled and masked version of the image. Rather than another saliency-to-saliency comparison, this experiment showed how humans vs. DNNs integrated features important to **humans** into object recognition decisions. Moreover, there was no “where” information at all in this experiment — all features were centered on the fovea. We have revised Section 4.2 to clarify.
>
> > Secondly, the authors make good use of this evaluation to show that improving ImageNet classification accuracy doesn't lead to better alignment with human saliency maps, underscoring the limitations of ImageNet as a benchmark. It would be interesting to see if other computer vision benchmarks, such as object detection or semantic segmentation are more predictive of human saliency.
>
> This is a great future direction, and we have added it to the Discussion.
>
> > Finally, it is encouraging to see that co-training the network to reproduce saliency maps allows it to do so, without any cost to the ImageNet classification accuracy (or even with a slight improvement to it), however it is unclear what the scientific value of this result is.
>
> We demonstrated that DNNs are systematically becoming **worse** models of human vision. We also introduced the Neural Harmonizer as a method for controlling this problem at scale. The scientific value here is identifying and fixing a fundamental flaw in DNNs, which has implications for theoretical and computational work in Neuroscience and Cognitive Science, and for industry in developing more reliable and predictable models of human visual behavior.
>
> > Aligning the training and testing objective usually improves test performance, hence improving the alignment with human saliency is hardly surprising.
>
> There is no guarantee that DNN alignment with humans could be improved on independent test sets from two different human feature importance experiments (*ClickMe* and *Clicktionary*), each of which relied on different methods, or that harmonizing a model would also improve its alignment with human decision making on our novel psychophysics (Figure 5). There is nothing trivial about these results and there’s no reason to discount them as unsurprising.

---

> ### Author Response · Authors · 2022-08-02
> **Continued**
>
> > A much more interesting point would be to investigate what training procedure, architecture, or data enables a model to reproduce human saliency without resorting to training on this data. This would indicate which of the above most closely matches the conditions leading to the human visual system's choice of features for performing these tasks…. Given the same model architecture and training data, which training procedure (supervised, self-supervised, etc) leads to the best alignment with human saliency? Given the same training data and objective (e.g. supervised or self-supervised), which model architecture (ResNet, ViT, etc) leads to the best alignment with human saliency? Given the same architecture and training objective, which training data (ImageNet, ImageNet21k, etc) leads to the best alignment with human saliency?
>
> The results you’re requesting are implicit in the performances of off-the-shelf models in Figures 3, 4, and 5 from the submitted manuscript. These models all use different augmentation strategies (e.g., adversarial training vs. the expansive augmentations of self-supervised models vs. the large-data training recipe of ViT), architectures, and training datasets. No model breaks the frontier described by the grey curves in Figures 3, 4, and 5 **except** for the harmonized models, which are significantly more aligned with humans and more accurate than their non-harmonized versions.
>
> The space of model parameterizations and training strategies is infinitely large and extremely hard and slow to search through. Moreover, there’s no guarantee that any neural network architecture can be expected to do as well as the Neural Harmonizer that we introduced in our manuscript, or that the best-performing configuration would tell us anything about brains or cognition.

---

> > ### Comment · Reviewer_xZSj · 2022-08-05
> > **From bad to worse**
> >
> > The authors seem to have misunderstood my comment.
> >
> > - The authors evaluate a host a models on their alignment with human saliency maps. They then start training networks for this same metric, and find that they improve in that regard. This is a trivial finding, and in their words, does not tell us "anything about brains or cognition."
> >
> > - Understanding brains and cognition means understanding why and how they have gained their particular selectivity and function. Evaluating which pretraining objective leads to most brain-like representations (as measured by the authors' proposed saliency alignment metric) would shed light on which learning objective the brain might be using. Note that the proposed "neural harmonizer" is not a candidate learning objective the brain might be using (this would be tautological), hence finding that the neural harmonizer outperforms other learning objectives teaches us nothing about what the brain is doing. It is simply hacking a metric that should be used for evaluation only.
> >
> > - Similarly, evaluating which neural network architecture leads to most brain-like representations (as measured by the authors' proposed saliency alignment metric) would elucidate which parametric class the brain might be using. However the authors simply lump all models together, confounding architecture, learning objective, and data. Hence we're unable to see any systematic effects from architecture. This is an unfortunately missed opportunity.
> >
> > Given that the neural harmonizer holds no scientific interest, and that the analysis section of the paper doesn't hold on its own (a more careful dissection of the results is required here), I don't think the paper belongs at NeurIPS.

---

> > > ### Author Response · Authors · 2022-08-05
> > > **Response**
> > >
> > > We thank the reviewer for their feedback
> > >
> > > > This is a trivial finding...
> > >
> > > To briefly restate our contributions:
> > >
> > > 1. In our survey of 85 neural networks trained for object classification, the best performing model explains just under 60% of human visual strategies.
> > > 2. Even more importantly, the alignment between DNNs and humans has worsened as their ImageNet accuracy has improved beyond the best-aligned model.
> > > 3. We show that harmonizing DNNs with human visual strategies improves their accuracy and significantly improves their alignment with humans.
> > >
> > > Reviewer *Xrpg* stated that our work "addresses an important problem in modern ML + CV."
> > > Reviewer *cZRY* stated that our work is "timely and relevant."
> > > Reviewer *cEc7* stated that we investigate an "extremely interesting problem," that "measuring if these improvements [in DNN accuracy] result in better alignment between biological and artificial intelligence is a fundamental goal for a multitude of reasons," and that "the neural harmonizer module is interesting as it increases alignment significantly."
> > >
> > > There is no guarantee that DNNs can learn to adopt human visual strategies in novel object images. We not only find that this is possible — across three datasets and two different measures of visual strategies (*where* humans look and *how* they integrate that information) — but also that this object is to some degree synergistic with object recognition accuracy. Our work therefore introduces a fundamental problem in computer vision, and shows that the solution is not only scalable but also useful for Brain Science and Industry applications alike since it both improves accuracy and promotes behavior that is more predictable and understandable to humans.
> > >
> > > > Evaluating which pretraining objective leads to most brain-like representations (as measured by the authors' proposed saliency alignment metric) would shed light on which learning objective the brain might be using...
> > >
> > > Sorry for the misunderstanding. We plotted the alignment of 85 DNNs trained for object recognition. No model exceeds 60% alignment with humans. In our original manuscript we stated that the best-aligned model is the DenseNet 121. Are you asking for a more detailed analysis of how these off-the-self models are ordered in their alignment?
> > >
> > > > Note that the proposed "neural harmonizer" is not a candidate learning objective the brain might be using (this would be tautological)...
> > >
> > > Our work is focused on DNNs as a model of behavior and their alignment with human visual strategies. We are not focused on biological plausibility since it is debatable that any DNN can pass that bar. Our use of the neural harmonizer is to show that co-training on human visual strategies surprisingly yields improvement in accuracy and predictions of human visual strategies on novel images.
> > >
> > > > Similarly, evaluating which neural network architecture leads to most brain-like representations (as measured by the authors' proposed saliency alignment metric) would elucidate which parametric class the brain might be using. However the authors simply lump all models together, confounding architecture, learning objective, and data. Hence we're unable to see any systematic effects from architecture.
> > >
> > > We clustered our analysis of 85 models according to whether they were standard CNNs trained on ImageNet, self-supervised, Transformers, adversarially-trained (Robust), trained with additional industry-level data (e.g., JIT-300M, JIT-3B, etc.), or harmonized. We did not spend much effort in the paper describing systematic differences between these architecture/training groupings because the factor that explained the most variance in human alignment was model accuracy. For instance, Transformers and models trained with extra data were less aligned with humans, but these are also the most accurate models that we tested.
> > >
> > > We are happy to add additional and more granular analyses. Could you please advise on what additional work you would like to see?

---

### Official Review · Reviewer_cZRY · 2022-07-11

**Rating:** 2
**Confidence:** 4
**Soundness:** 1 poor
**Presentation:** 3 good
**Contribution:** 2 fair

**Summary:**

The authors address the problem of evaluating the alignment of deep learning vision models with humans. To do so, they compare feature importance maps by humans, with saliency maps from DNNs. Additionally, they propose a training method (neural harmonizer) to improve such alignment. Finally authors present a psychophysic experiment using the rapid categorization paradigm to gather information of human decisions based on partial information of important features.

**Questions:**

I would be incline to change ma opinion if authors:
- provide strong motivations on the choice of the one method for saliency, or present results for at least another different approach.
- provide strong motivations on the choice of the two models to be improved with neural harmonizer, or provide average results for all initial 85 models
- clarify why using feature importance map is sufficient to explain and measure alignment, in particular w.r.t. research in saliency and scanpath that rely directly on eye-tracking data

**Limitations:**

The discussion of limitations is insufficient. The authors should point out that results are only provided for one explainability method, and only for two models. Additionally, they should point out how the notion of feature importance and what is actually captured by humans through the gaze might differ.

**Strengths And Weaknesses:**

Strength:
- the topic is timely and relevant
- the paper is clear and easy to read
- the authors validate part of their claims with a human study
- code is provided for reproducibility

Weaknesses:
- Structure of the paper is not optimal. The neural harmonizer should probably be described introduced in the method section.
- The rapid categorization paradigm has already been used to compare humans and DNNs decisions and it could be beneficial to provide those references (the provided references are from the neurosc. field)
- Author exclusively use saliency maps from Simonyan et al. to reveal feature importance. However, studies have found severe weaknesses of such metrics that can also be drastically attacked. Authors could provide more arguments on why they chose to rely specifically and exclusively on this method, or eventually use more.
- Related to the previous point, the proposed neural harmonizer method optimises neural networks to change their feature importance (w.r.t. the selected method of saliency). I rather have a different interpretation than those provided by the authors on the effect of this approach: this might lead the network to learn more effective activation that lead to human-like feature importance (maybe leveraging spatial biases of human annotators) while not changing the underlying decision process. The improvement in accuracy might be due to the additional loss and training data (in fact, multi-tasking is performed). Which lead me to the following observation:
- Why do authors provide results for only two networks (ResNet50 and VGG16) of the 85 presented in the first experiment, and why those two? Their explanation is given in page 6, they are "selected because they are popular networks that sit on the boundary of the trade-off betweenDNN performance and alignment with humans". This seems to be not fully true from figure 4. Additionally, it might be very useful to know the average performance on the other architectures (or at least a random sample of them, if training them all is prohibitive) in order to support the claim that the method improves alignment without decreasing the performance. Especially important are the performance on DenseNet121, mentioned by the authors as the very limit of the trade-off between alignment and accuracy.
- Authors assume that feature importance maps from the two selected datasets represent where features which are important for the decisions are in a scene. However, I am not sure this aligns with locations people attend with their gaze. Authors should discuss this point in the paper. Eventually, an experiment involving saliency maps (using for example SALICON, a subset of imagenet with human visual saliency information) might be meaningful?

Minor:
- Page 4, row 2, "was was" (repetition)
- The authors refer to feature maps also as "visual strategies" often during the paper, however, I found this to be confusing since in the field this is more often used for visual scanpaths than for (static) saliency. I suggest use a different terminology.
- I am not sure wether presenting a partial image cropped is effective since, even when selecting a smaller area, people can see the rest of the image with peripheral vision. Trying to model this effect could have made the experiment more reliable.
- Section 4.1: motivate the use of spearman's rank-correlation or provide references
- In the paragraph "The neural harmonizer aligns human and DNN visual strategies.": authors mention "All of the three neural harmonizer models...", however they presented only two of them.

---

> ### Author Response · Authors · 2022-08-02
> **Response**
>
> > The rapid categorization paradigm has already been used to compare humans and DNNs decisions and it could be beneficial to provide those references (the provided references are from the neurosc. field)
>
> We cited Eberardt et al., 2016, Linsley et al., 2018, and Serre T., 2007 in our original submission, which describe experiments comparing humans to models using rapid visual categorization. Are there any others you have in mind? We are happy to add more that we missed that fit the scope of our paper.
>
> > Author exclusively use saliency maps from Simonyan et al. to reveal feature importance. However, studies have found severe weaknesses of such metrics that can also be drastically attacked. Authors could provide more arguments on why they chose to rely specifically and exclusively on this method, or eventually use more.
>
> As we discuss in our Response to all reviewers, saliency was chosen because it is fast to compute, and we found with it some models are nearly perfectly aligned with humans while others are not at all aligned with humans. We have added SI Figure 11 ([also see here](https://drive.internxt.com/s/file/1dd1d40e0a2b70243344/7bc64eb592e94dcb697779acb26e4549511fa4ba0bc4ace83c1ba73fea180f5b)) showing that our findings replicate when testing models with SmoothGrad [1] instead.
>
> > Related to the previous point, the proposed neural harmonizer method optimises neural networks to change their feature importance (w.r.t. the selected method of saliency). I rather have a different interpretation than those provided by the authors on the effect of this approach: this might lead the network to learn more effective activation that lead to human-like feature importance (maybe leveraging spatial biases of human annotators) while not changing the underlying decision process.
>
> Can you clarify what you mean by “more effective activation?” The average of *ClickMe* maps used for training yields a center bias, which explains 42% of human variance, which is significantly worse than 21 unharmonized DNNs and all of our harmonized models. Our psychophysics experiments in Figure 5 demonstrate that harmonizing models indeed changes their decision making, and how they integrate visual features into decisions, to be more like human observers.
>
> > The improvement in accuracy might be due to the additional loss and training data (in fact, multi-tasking is performed).
>
> There’s no guarantee that co-training on human feature importance maps from *ClickMe* should preserve or improve accuracy. Our findings show that the two objectives are at least partially synergistic in large-scale training routines.
>
> > Why do authors provide results for only two networks (ResNet50 and VGG16) of the 85 presented in the first experiment, and why those two?
>
> See point 1 in **Response to all reviewers** for our explanation and new modeling results ([Figure 3](HYPERLINK), [Figure 4](HYPERLINK), and [Figure 5](HYPERLINK)). We assure you we are not cherry picking results. Instead, the compute needed to harmonize each of the 85 models that we tested was more than we had available in our academic lab. We include significance tests in our revision showing that the pareto front represented by the harmonized models is significantly higher than the unharmonized models.

---

> ### Author Response · Authors · 2022-08-02
> **Continued**
>
>
> > Authors assume that feature importance maps from the two selected datasets represent where features which are important for the decisions are in a scene. However, I am not sure this aligns with locations people attend with their gaze. Authors should discuss this point in the paper. Eventually, an experiment involving saliency maps (using for example SALICON, a subset of imagenet with human visual saliency information) might be meaningful?
>
> The main distinction to make is in whether human feature importance maps reflect bottom-up or top-down Saliency. SALICON reflects bottom-up saliency, and aligns with human eye-tracking data collected when participants are asked to look at images without any defined task. In the original *ClickMe* paper [1], they proposed that the *ClickMe* dataset instead reflects top-down, or task-driven saliency. They demonstrated the difference between bottom-up (SALICON) and top-down (*ClickMe*) saliency by asking individuals to categorize object images when only a proportion of the most important features were visible. In one case, features were revealed according to SALICON importance, and in the other case features were revealed according to *ClickMe* importance. Human participants were significantly more accurate at recognizing objects when features were revealed according to *ClickMe*, even when only ~10% of those features were visible.
>
> Our psychophysics experiments in Figure 5 support the finding that *ClickMe* captures top-down and task-driven saliency. DNNs that were harmonized with *ClickMe* feature importance were also significantly more aligned with human decision-making processes. It is always possible that an extremely controlled eye tracking study could produce data that are a more reliable indicator of human decision-making processes, and the Neural Harmonizer could certainly be used with that data, but there is no eye tracking dataset that is anywhere near large enough for co-training DNNs.
>
> > The authors refer to feature maps also as "visual strategies" often during the paper, however, I found this to be confusing since in the field this is more often used for visual scanpaths than for (static) saliency. I suggest use a different terminology.
>
> We have added a sentence to the methods to clarify this. *ClickMe* and *Clicktionary* maps are the average of mouse scanpaths — participants paint the parts of the object they believe are important for classifying it. In that sense it is similar to your definition, albeit far less rich.
>
> > I am not sure wether presenting a partial image cropped is effective since, even when selecting a smaller area, people can see the rest of the image with peripheral vision. Trying to model this effect could have made the experiment more reliable.
>
> See Figure SI 1 (included in the original submission) for the psychophysics method and more examples of how we revealed only a proportion of the most human-important features to machines and human participants. There is no diagnostic information in the periphery. Please let us know if we’ve misunderstood your comment.
>
> > Section 4.1: motivate the use of spearman's rank-correlation or provide references
>
> We added a reference for this to the Results section where we introduce Spearman’s rank-correlation.

---

### Official Review · Reviewer_Xrpg · 2022-07-11

**Rating:** 4
**Confidence:** 5
**Soundness:** 2 fair
**Presentation:** 3 good
**Contribution:** 2 fair

**Summary:**

This paper re-introduces a very important topic in the ML + CV community that has not been studied enough: are computer vision models aligned to human visual perception (and if they are not, why should we care?). The authors introduce the neural harmonizer as a training procedure that aligns the human-relevant features with a CNN to train from scratch a couple of new models from classical CNN architectures (ResNet + VGG) and they show that their models improve human perceptual alignment while also maintaining ImageNet Test Accuracy and performance. Furthermore, they empirically show that networks trained with the Neural Harmonizer are more robust to a peripheral occlusion task, and most importantly -- that their change in performance rate is similar to that of a human without any additional fine-tuning on the network.

**Questions:**

See Weaknesses points (1) and (3).

**Limitations:**

None, the paper actually does a great job in introducing current limitations of modern DL systems. The paper is overall well written. Providing additional experiments and/or a good rebuttal may improve my rating. I am also generally curious to know what other reviewers have thought about the maturity of this paper.

I'm really on the fence with this paper. It's amazingly well written, addresses an important problem in modern ML + CV -- via the proper cog-sci/psychology insights, but is still missing more results that back-up their claim with stronger evidence. There are no error bars for example, and authors could have definitely trained the other NN's with their Neural Harmonizer, to provide substantial evidence about how the Harmonizer provides better human-perceptual alignment.

**Strengths And Weaknesses:**

The paper’s main strengths is the paper’s theme as well as the very accessible way the paper has been written. Furthermore the paper’s experiments seem properly articulated and well-posed to support the main hypothesis and question that the authors introduce at the beginning of the paper. The psychophysics is pretty amazing.

However, the paper still has some weaknesses that should be addressed/clarified:
* **Weak Results:** Why are there only 2 reported neural networks that have been trained with the neural harmonizer (there are 2 purple dots)? I would have expected that for the claim to be “the neural harmonizer successfully increases man-machine perceptual alignment”, that there would be nearly all blue CNN dots transcribed to purple, including (potentially) the yellow ones which are the Transformers. Without these extra data points, the results seem very weak, and almost suggests cherry picking.
* **Missing References:**
    * “Emergent Properties of Foveated Perceptual Systems”. Deza & Konkle. ArXiv, 2020.
    * “Central and peripheral vision for scene recognition: A neurocomputational modeling exploration” Wang & Cottrell. JoV, 2017.
    * “Joint rotational invariance and adversarial training of a dual-stream Transformer yields state of the art Brain-Score for Area V4”. Berrios & Deza. Brain-Score Workshop 2022.
    * “Texture-like representation of objects in human visual cortex”. Jagadeesh & Gardner. PNAS 2022.
* **Experimental Results may provide evidence for alternative explanation**: The occlusion results from Figure 5 may also suggest that the neural harmonizer has enforced a central image bias in CNNs :O ! (So a partial rather than full human-perceptual alignment). It is expected that Transformers would not a priori have this bias. That being said: How would the Transformer’s curve trained with a Neural Harmonizer look like? In addition, what would the curve look like if the occlusion patterns were inverted, and one where to begin by starting to show peripheral regions of the image, and slowly showing the most important features last? Do the same patterns of results hold between humans and neural harmonizer trained networks? See for example the work of Deza & Konkle (ArXiv, 2020) and also Wang & Cottrell (JoV, 2017)

---

> ### Author Response · Authors · 2022-08-02
> **Response**
>
> > *Weak Results*: Why are there only 2 reported neural networks that have been trained with the neural harmonizer (there are 2 purple dots)? I would have expected that for the claim to be “the neural harmonizer successfully increases man-machine perceptual alignment”, that there would be nearly all blue CNN dots transcribed to purple, including (potentially) the yellow ones which are the Transformers. Without these extra data points, the results seem very weak, and almost suggests cherry picking.
>
> See point 1 in **Response to all reviewers** for our explanation and new modeling results ([Figure 3](https://drive.internxt.com/s/file/53eb82c0084e2ec814bc/371fefa1acbf6aea724abed91895a4ba4eefafb469cc68c67cf2291b7df73740), [Figure 4](https://drive.internxt.com/s/file/9ff6afa4279f8049163f/c87fc2a992ea72ddc5e066c2f9f8c7c910ca7e6259cf23b80b597773b9baa904), and [Figure 5](https://drive.internxt.com/s/file/aa6ce7313067acf1b50e/acdc4983f07811e7859df6bf4f246a79e36e22ef06e07610e3bac6299b798ae5)). We assure you this is not cherry picking; rather, the compute needed to harmonize all 85 models that we tested was more than we had available in our academic lab.
>
> > That being said: How would the Transformer’s curve trained with a Neural Harmonizer look like?
>
> This model is non-trivial to harmonize because of the large datasets used to train it. We have revised our discussion to discuss how harmonizing Visual Transformers and other models trained on ultra-large datasets used in industry groups (e.g., JIT-300M or JIT-3B) is a future direction of the work.
>
> > *Experimental Results may provide evidence for alternative explanation*: The occlusion results from Figure 5 may also suggest that the neural harmonizer has enforced a central image bias in CNNs :O ! (So a partial rather than full human-perceptual alignment).
>
> This is an extremely insightful and smart suggestion. *ClickMe* does indeed have a center bias when all maps are averaged together ([click here to see the average map](https://drive.internxt.com/s/file/16c375747819dab9f369/2da19b5baaa28bf200852732d0f383c25eebe6040811930421bfd96320150c49))! A center bias is fundamental in any measure of saliency, such as eye-tracking, and hence it is important to understand what a saliency dataset offers above-and-beyond such a trivial bias. That said, this bias raises the unfortunate possibility that collecting *ClickMe* maps is a waste of time; all you need to do is co-train models on a centered gaussian to align them with humans.
>
> To test this possibility, we measured the alignment between the center bias computed from all *ClickMe* maps and the *ClickMe* maps for each individual image. We found that the center bias explained 42% of human variance, which is a surprisingly high amount but also significantly lower than 21 unharmonized DNNs and all four of our harmonized models. See the cross in Figure 3 for the marker corresponding to the *ClickMe* center bias.
>
> Inspection of individual *ClickMe* maps shows that they mostly focus on parts of foreground objects, such as the faces of animals or fronts and wheels of vehicles (SI Figure S3), thus it is likely that the surprising ability of a center bias to explain 42% of human variance in *ClickMe* maps is a function of where objects typically are positioned in ImageNet images rather than a center spatial bias per se as is discussed in the context of saliency and eye tracking.
>
> Notably, the largest-scale models (e.g., ConvNext, ViT, and CLIP) were **less** aligned with humans than this center bias, indicating that they have high inter-image variability in the features they find important for recognition and that these features are systematically out-of-alignment with humans.
>
> We have added the citations you suggested and a discussion of this center bias in the results section of the main text.
>
> >It is expected that Transformers would not a priori have this [center] bias.
>
> Could you please clarify this comment? Why would any of the models we tested (Transformers or any of the CNNs) have an inductive bias to focus on the center of the image?
>
> >How would the Transformer’s curve trained with a Neural Harmonizer look like?
>
> We have expanded our discussion to include Transformers and other ultra-large scales models, and issues involved with harmonizing them. In short, models which were trained with industry datasets like JIT-300M present a challenge for harmonization because of the computational resources they require. Furthermore, these datasets may need additional feature importance maps from humans (via *ClickMe* for example) in order to properly harmonize.

---

> > ### Comment · Reviewer_Xrpg · 2022-08-09
> > **Preserving Score. Next iteration of paper may be ready**
> >
> > Given the feedback from all reviews and the rebuttal, I've decided to keep my score. I think the paper is making progress but it is still not ready for publication. In addition I find this claim in the general rebuttal quite strong:
> >
> > > "Lastly, we want to re-emphasize that we have provided the first ever demonstration that the field of computer vision is systematically producing worse models for Human vision."
> >
> > I do not think this is true. The work that Schrimpf has pioneered through Brain-Score and through a series of papers has already shown this. Geirhos has also in parallel explored many of these pivotal  ideas via rigorous human vs machine psychophysics and several other results (texture-bias, short cuts, etc...) . The Neural Harmonizer and/or the general structure of the paper should give a greater insight beyond what Schrimpf, Geirhos and other colleagues that perform work at the intersecion of human and machine vision have explored.

---

> > > ### Author Response · Authors · 2022-08-10
> > > **Response**
> > >
> > > We thank the reviewer for their response and second look at the paper.
> > >
> > > We would like to respond to your comment on the quoted claim.
> > >
> > > Brain Score describes the performance of models trained to *predict neural activity recorded in primates, not humans*. In our work (and the quote) we are solely comparing DNNs with human perception. Also note that in our submitted manuscript we describe how Brain Score has changed over the years as DNNs have become more accurate (lines 82-84): "While the original theory was that model explanations of neural representations of object images improved alongside model classification accuracy [29], recent large-scale DNNs are worse at explaining neural data than older ones with lower ImageNet accuracy [30]."
> > >
> > > Geirhos indeed has done rigorous work comparing human and DNN classification performance on a variety of challenging tasks. His most recent paper, "Partial success in closing the gap between human and machine vision" aggregates most of the prior challenges together. In our original submission we summarized their findings (lines 69-73), "On the one hand, there is evidence that DNNs are becoming better models of humans on challenging visual tasks, such as recognizing objects obscured by noise [23]." The authors found that newer models like CLIP perform nearly on par with humans, whereas we found that CLIP is poorly aligned with human perception and visual strategies.
> > >
> > > Given the above, we hope you agree that we do indeed provide novel insight beyond Schrimpf and Geirhos. Please let us know if you have any other references that we have missed, we would be happy to add them.

---

> ### Author Response · Authors · 2022-08-02
> **Continued**
>
>
> > In addition, what would the curve look like if the occlusion patterns were inverted, and one where to begin by starting to show peripheral regions of the image, and slowly showing the most important features last? Do the same patterns of results hold between humans and neural harmonizer trained networks? See for example the work of Deza & Konkle (ArXiv, 2020) and also Wang & Cottrell (JoV, 2017)
>
> The *Clicktionary* maps we used to sample important object features in for the Figure 5 psychophysics are used in total — the X-axis depicts performance for 1% of the most important features to 100%.
>
> Are you asking for a comparison of object recognition performance between humans and DNNs on features which were not included in ClickMe maps? Do you have any advice on how to sample those features in a way that could be comparable to our psychophysics experiments?
>
> One piece of evidence from the original *ClickMe* paper that may be relevant is that they compared human classification accuracy for objects masked to only reveal *ClickMe* features to objects masked to only reveal features that were salient according to the SALICON dataset. Participants were significantly more accurate at recognizing objects revealing *ClickMe* features vs. SALICON features.
>
> > There are no error bars for example, and authors could have definitely trained the other NN's with their Neural Harmonizer, to provide substantial evidence about how the Harmonizer provides better human-perceptual alignment.
>
> Please see SI Fig. 6 (included in our original submission) for the performance of many different harmonized ResNet50s. We have also added error bars for human alignment and statistical tests to the main text. Thank you for this suggestion (in addition to all of the others!!), which reinforces and strengthens our findings.

---

### Author Response · Authors · 2022-08-02
**Response to all reviewers**

Thank you to the reviewers for taking the time to read and review our paper. Your critiques are sharp and insightful and we found that all four reviewers converged on the following two issues which we have addressed over the rebuttal period:

1. Why did we only include two harmonized DNNs in our manuscript?
2. Why does the paper focus on gradient-based saliency for comparing humans and model feature maps and for training our Neural Harmonizer?

We have addressed these issues in the following ways:

1. (**Xrpg**, **cZRY**, ) We focused on VGG16 and ResNet50 in our original version of the manuscript because they are featured in nearly all Neuro/Cog Science research that uses DNNs, and essentially every CNN architecture is derived from these two. We believed that if we were to show improvements in models that include/don’t include residual connections, then we were confident in the effectiveness of our Neural Harmonizer method.

+ That said, we have taken to heart the Reviewers’ suggestion and over this review period we have harmonized two more CNNs, the MobileNet and EfficientNet_b0. Like the VGG16 and ResNet50 we discussed in our submission, both of these harmonized models improved over their unharmonized versions in top-1 accuracy and became significantly more aligned with human visual strategies for object recognition. Furthermore, following a suggestion from reviewer **Xrpg**, we found that the pareto front describing our harmonized models allowed for significantly better accuracy and alignment with humans than the pareto front describing non-harmonized models. See [Figure 3](https://drive.internxt.com/s/file/53eb82c0084e2ec814bc/371fefa1acbf6aea724abed91895a4ba4eefafb469cc68c67cf2291b7df73740), [Figure 4](https://drive.internxt.com/s/file/9ff6afa4279f8049163f/c87fc2a992ea72ddc5e066c2f9f8c7c910ca7e6259cf23b80b597773b9baa904), and [Figure 5](https://drive.internxt.com/s/file/aa6ce7313067acf1b50e/acdc4983f07811e7859df6bf4f246a79e36e22ef06e07610e3bac6299b798ae5) in the main text (or at each respective hyperlink) to find the additional results.

2. ( **cZRY**) We focused on saliency in the paper for two reasons. First, we designed the neural harmonizer to scale with the size of state-of-the-art DNNs for object recognition. That means that we sought to minimize the additional compute needed to harmonize models. Saliency is the computationally cheapest model explanation to calculate. Second, we found that the saliency maps from some models (harmonized VGG-16) had a nearly perfect correlation with human feature maps, whereas other models (ConvNext) had extremely low correlations with human feature maps. Visual inspection (Figure 2 in the main text) of these maps reinforces this insight that existing DNNs mostly rely on features in different spatial locations than humans to make their object recognition decisions and that harmonization fixes this problem.

+ As a check on whether or not the features revealed by saliency are idiosyncratic to that method, we recomputed human feature alignment on the *ClickMe* test dataset using SmoothGrad instead, which has been described as the most “reliable” feature visualization method [1]. See SI Figure 11 (or [here](https://drive.internxt.com/s/file/1dd1d40e0a2b70243344/7bc64eb592e94dcb697779acb26e4549511fa4ba0bc4ace83c1ba73fea180f5b)) for these results, which show the same trade-off and correction induced by the Neural Harmonizer that we describe for saliency in the main text.

In addition to the above results, over this review period, we have added citations and clarifications to our Related Work and Discussion (**Xrpg**, **cEc7**, **cZRY**), added an additional center bias control which validates that effectiveness of our Neural Harmonizer, and included error bars and statistical tests wherever possible  (**Xrpg**). We will address each reviewer’s remaining comments directly.

Lastly, we want to re-emphasize that we have provided the first ever demonstration that the field of computer vision is systematically producing worse models for Human vision. This finding is enormously consequential for Neuroscience, Cognitive Science, and industry alike, each of which rely on neural network architectures to predict human behavior and visual representations. Our Neural Harmonizer is an efficient drop-in solution that can rectify this problem in large-scale neural network models. We thank the reviewers for their suggestions which have bolstered the evidence for our findings.

[1] Colin J, Fel T, Cadene R, & Serre T. 2022. What I Cannot Predict, I Do Not Understand: A Human-Centered Evaluation Framework for Explainability Methods. ArXiv.

---

> ### Author Response · Authors · 2022-08-08
> **A note before the end of the response period on 8/9**
>
> Dear reviewers,
>
> Thank you again for your time and effort in reading and critiquing our paper. We hope you will have a chance to read our responses to your reviews before the response period ends tomorrow, 8/9. Please find that we have addressed the concerns raised, and that if you have any others we are happy to discuss them.
>
> Best,
>
> The authors

---

### Meta-Review · Area_Chair_bq26 · 2022-08-27

**Recommendation:** Accept
**Confidence:** Less certain

**Metareview:**

The reviewers have brought up important concerns around the framing of the paper contributions, the presentation and application of the neural harmonizer method, and the use of saliency maps. The authors have addressed some of these concerns in their rebuttal.

However, I think the main contribution of this paper is the evaluation study itself, which examines 85 different neural network architectures and compares the important features for those architectures (via saliency maps) to the features humans consider important, for varying levels of performance.

This topic of human vision and computer vision inductive biases has been studied through e.g. comparisons of shape vs texture bias, but I think the study presented in this paper (Spearman correlation between feature importance maps) is an interesting addition to work in this area and has scope to be built on by the community, helped by the code release.

I do think the authors could have explored additional empirical studies around this main point, e.g. comparing saliency approaches to self-attention for vision transformers, or looking at the effect of architecture scale on salient features and comparison to human salient features. This to me is the main gap of the paper.

The paper is thus borderline, with an interesting main study contribution, with scope to be built on and a code release, but lacking additional related empirical experiments.

**Award:**

No

---

### Decision · Program_Chairs · 2022-09-14

Accept